# The small CRL4^CSA ubiquitin ligase component DDA1 regulates transcription-coupled repair dynamics

Diana A. Llerena Schiffmacher [1,15], Shun-Hsiao Lee [2,15], Katarzyna W. Kliza [3,13], Arjan F. Theil [1], Masaki Akita [1,14], Angela Helfricht[1], Karel Bezstarosti[4], Camila Gonzalo-Hansen[5], Haico van Attikum[6], Matty Verlaan-de Vries[7], Alfred C. O. Vertegaal[7], Jan H. J. Hoeijmakers [1,8,9], Jurgen A. Marteijn [5], Hannes Lans [1], Jeroen A. A. Demmers [4], Michiel Vermeulen [3,10], Titia K. Sixma [2], Tomoo Ogi [11,12], Wim Vermeulen [1] ✉ & Alex Pines [1] ✉

Transcription-blocking DNA lesions are specifically targeted by transcription-coupled nucleotide excision repair (TC-NER), which removes a broad spectrum of DNA lesions to preserve transcriptional output and thereby cellular homeostasis to counteract aging. TC-NER is initiated by the stalling of RNA polymerase II at DNA lesions, which triggers the assembly of the TC-NER-specific proteins CSA, CSB and UVSSA. CSA, a WD40-repeat containing protein, is the substrate receptor subunit of a cullin-RING ubiquitin ligase complex composed of DDB1, CUL4A/B and RBX1 (CRL4^CSA). Although ubiquitination of several TC-NER proteins by CRL4^CSA has been reported, it is still unknown how this complex is regulated. To unravel the dynamic molecular interactions and the regulation of this complex, we apply a single-step protein-complex isolation coupled to mass spectrometry analysis and identified DDA1 as a CSA interacting protein. Cryo-EM analysis shows that DDA1 is an integral component of the CRL4^CSA complex. Functional analysis reveals that DDA1 coordinates ubiquitination dynamics during TC-NER and is required for efficient turnover and progression of this process.

Different DNA damage repair mechanisms and signaling pathways collectively preserve genome stability, protect cells against DNA-damaging agents and are key to maintaining proper cellular functioning and thereby counteract both carcinogenesis and aging[1,2]. Among the different DNA repair systems, nucleotide excision repair (NER) stands out for its versatility in removing a broad spectrum of base-pair-disturbing DNA lesions through an intricate multistep process[3–5]. NER removes lesions by two complementary sub-pathways: Global Genome NER (GG-NER), which repairs helix-distorting DNA damage throughout the whole genome, and Transcription-Coupled NER (TC-NER), which removes transcription-blocking DNA lesions

(TBLs) in the transcribed strand of active genes and preserves thereby the crucial transcription programs. After damage recognition, the two sub-pathways share the downstream steps. The transcription factor IIH (TFIIH) binds to DNA and, using its translocase and helicase activity opens up the DNA and checks for the presence of DNA damage[6]. This central activity leads to excision of the lesion-containing single-stranded DNA, followed by restoration of the strand by DNA synthesis and ligation[3–5].

TC-NER is initiated by stalling of elongating RNA polymerase II (RNAPII) onto DNA lesions[7,8], which triggers the binding of Cockayne syndrome B protein (CSB), a member of the SWI2/SNF2 family[9]. While,

---

CSB normally interacts transiently with RNAPII for monitoring progression and to facilitate the translocation over intrinsic pausing sites and smaller lesions, upon encountering helix-disturbing TBLs, CSB becomes firmly bound to RNAPII[8] and enables further assembly of the other TC-NER core factors CSA and UVSSA[3–5]. CSA is a WD40 domain-containing protein that belongs to the DCAF (DDB1- and CUL-associated factors) family[10] and is the substrate receptor subunit of the DDB1, CUL4A/B and RBX1 containing cullin-RING ubiquitin ligase complex (CRL4^CSA)[11,12]. The activity of this ligase is activated by the NEDD8 conjugation to CUL4 and negatively regulated by the COP9 signalosome (CSN) complex[13]. Upon UV-light-induced DNA damage, CRL4^CSA gets activated and ubiquitinates different TC-NER-associated substrates, including RNAPII[14], UVSSA[15] and CSB[16]. The CSA-dependent CSB ubiquitination is counteracted by the broad-spectrum de-ubiquitinating enzyme USP7 which is recruited to the TC-NER complex by UVSSA[17,18]. UVSSA itself is also ubiquitinated at Lysine 414, to facilitate TFIIH recruitment and promote RNAPII ubiquitination[14,15,19]. Additionally, previous studies have also shown that other ubiquitin ligases and ubiquitin-chain editing enzymes are implicated in differential RNAPII ubiquitination[4]. These complex and dynamic ubiquitination events on RNAPII were proposed to determine the fate of lesion-stalled RNAPII, which either drive the timely association and likely also dissociation of the TC-NER factors, or are implicated in the removal and degradation of lesion-stalled RNAPII, or control genome-wide dissociation of promotor-paused RNAPII in response to TBLs[14,15,20]. Recently, the structure of human elongating RNAPII bound by CSB, CRL4^CSA and UVSSA was resolved using cryo-electron microscopy (cryo-EM) and has provided important structural information on how the elongating RNAPII complex is converted into a TC-NER complex which forms the basis for coupling transcription to DNA repair in human cells[21].

Although CSA has been studied extensively for many years and valuable insight into the molecular interactions and possible ubiquitination targets have been obtained, we still know very little about how the CRL4^CSA is controlled and interconnected with the repair machinery at DNA damage-stalled RNAPII. To study this, we applied a MS approach using a fluorescently tagged CSA knock-in (KI) cell line and identified DDA1[22] as an important binding partner of CSA. DDA1 is a core subunit of multiple, though not all, Cul4-based E3 ligases[23]. We showed that DDA1 is an integral component of the CRL4^CSA complex and that it coordinates the CRL4^CSA complex activity and facilitates TC-NER progression. Our findings suggest that not only a highly controlled cooperative assembly but also a timely turnover of TC-NER proteins is important in regulating the progression of DNA repair to preserve the transcription program integrity.

## Results

### Generation and characterization of CSA-mClover knock-in cells
How CRL4^CSA operates and controls TC-NER is presently a matter of debate[14,20]. Further information on CRL4^CSA functioning might be provided by defining the composition of this protein complex. To comprehensively chart the CSA interactome in human cells, we applied biochemical techniques in concert with quantitative mass spectrometry (MS)[24] using Stable Isotope Labeling by Amino acids in Cell culture (SILAC)[25]. To efficiently isolate CSA-containing complexes, we generated a homozygous CSA knock-in HCT116 cell line by CRISPR-Cas9-mediated genome editing[26], that expresses a fluorescently tagged version of endogenous CSA. mClover DNA, a modified version of GFP[27], was inserted at the 3′ end of the *CSA* gene. Sequencing confirmed the proper in-frame integration of the mClover tag, and immuno-blot analysis showed that CSA-mClover is expressed at equal levels as non-tagged CSA in the parental cell line (Supplementary Fig. 1A, B). Colony Survival, revealed that the CSA-mClover knock-in cells (hereafter CSA-mC KI) were equally resistant to UV-C light (hereafter UV)-induced DNA damage as the cognate parental wild-type HCT116 cells, in contrast to the highly UV-sensitive CSA HCT116 knock-

out cells (Supplementary Fig. 1C). Moreover, Recovery of RNA Synthesis[28] (RRS) analysis by 5-EU labeling after UV irradiation (Supplementary Fig. 1D), showed that the TC-NER activity is not affected by the mClover tag. To further validate the CSA-mC KI cell line, we performed fluorescence redistribution after photobleaching[29] (FRAP). In this assay, fluorescent proteins are photobleached in a narrow strip spanning the cell nucleus by a high-intensity laser pulse. The subsequent fluorescence redistribution is monitored in time, providing a measure for the protein's mobility and dissecting different kinetic pools, e.g., free diffusing and/or chromatin-bound fractions in living cells. FRAP experiments showed a markedly reduced fluorescence recovery of CSA-mClover after 10 J m$^{-2}$ of UV irradiation, indicative of binding (immobilization) of CSA-mClover to chromatin-bound lesion-stalled RNAPII (Supplementary Fig. 1E). This UV-induced immobilization was absent in presence of transcription inhibitor THZ1. Importantly, UV-induced CSA-mClover immobilization is completely reverted to the pre-UV-damaged situation 10 h after UV (Supplementary Fig. 1F). Both the transcription dependency and reversion of immobilized CSA-mClover 10 h after UV, when most TBLs are repaired, clearly indicate that this immobilization reflects active participation in TC-NER. Together these TC-NER activity assays demonstrate that the generated CSA-mC KI cell line is a bona fide and highly sensitive tool to study the binding kinetics of CSA in TC-NER and provides a valid source to capture CSA-associated proteins.

### MS analysis revealed DDA1 as an interaction partner of CSA
Here we used the mClover (GFP derivative) tag as bait for affinity purification to isolate protein complexes by a simple, single-step affinity purification protocol employing GFP-Trap beads[30]. CSA-mC KI cells were mock treated or irradiated with UV and the CSA-mClover-containing protein complex(es) were isolated by immunoprecipitation and identified by LC-MS. This methodology is based on stringent washing conditions coupled with highly selective and specific GFP-bead purification to obtain stable complexes of significant purity for MS analysis[31,32]. SILAC analysis under non-damaging conditions, comparing CSA-mC KI with the parental HCT116 cells, identified known CSA-specific interacting proteins[12]. These include the CRL complex subunits, all subunits of the COP9 signalosome and the chaperonin complex TRiC, previously shown to be essential for proper CSA incorporation into the CRL complex[33] (Fig. 1A).

Surprisingly, CSB appeared already associated with CRL4^CSA even in the absence of exogenously induced TBLs, suggesting that either a fraction of CSA and CSB are already connected prior to their binding to lesion-staled RNAPII or that in non-UV-challenged cells TC-NER is continuously active towards endogenously induced TBLs. We similarly examined the changes in the CSA interactome upon UV irradiation by comparing mock-treated and UV-treated CSA-mC KI cells. Numerous previously described TC-NER interacting proteins were identified[34,35], including several RNAPII subunits and RNAPII-associated factors, PAF-1 complex and the TFIIH complex, validating the screen (Fig. 1B). In addition, we identified the DET1 and DDB1 Associated protein 1 (DDA1) as a constitutive interacting component of CRL4^CSA (Fig. 1A, B and supplementary Data 1) both in the presence and absence of UV-induced DNA damage. Immuno-blot analysis of CSA-mClover pull-downs from both CSA-mC KI and CSA-GFP overexpressing cell lines confirmed the interaction of CSA with the CRL (CUL4A and DDB1), COP9 (CSN5), TC-NER (CSB), and the TRiC (TCP-1) complex and validated the specific interaction between CSA and DDA1 (Fig. 1C and Supplementary Fig. 2A). DDA1 was previously identified as a subunit of several CRL4-E3 ligase complexes[23], but remarkably was not found associated with the CRL4^DDB2, which is involved in DNA damage recognition within GG-NER, and is structurally comparable to CRL4^CSA [11]. To further investigate the selectivity for CRL4^CSA, we generated GFP-DDB2 HCT116 knock-in cells by inserting a GFP tag at the N-terminus of DDB2 locus. These cells were validated by immune-blot

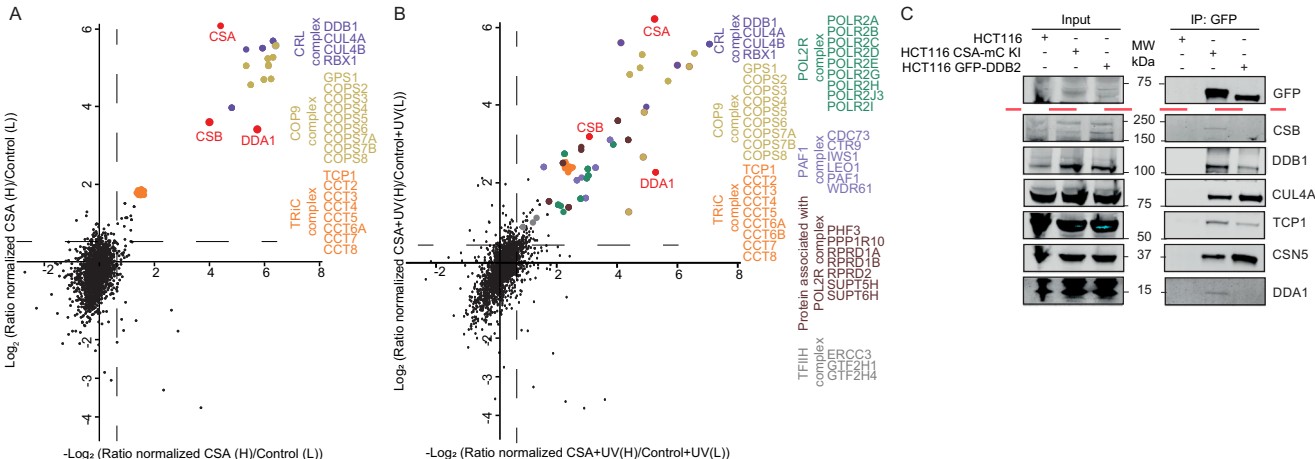

**Fig. 1 | DDA1 is an interaction partner of CSA. A, B** Scatter plot of Log$_2$ SILAC ratios of proteins isolated by GFP-pulldown in CSA-mC KI HCT116 cells. The experiments were conducted in duplicate with a label swap, comparing the GFP immunoprecipitation of mock-treated CSA-mC KI versus HCT116 cells (**A**) or UV-treated CSA-mC KI versus HCT116 cells (**B**). Proteins with Log$_2$ SILAC ratio >0.6 (indicated by grey line) in both replicates were classified as specific CSA interactors. RNAPII subunits are indicated in green, PAF-1 subunits are indicated in light purple,

proteins associated with RNAPII are indicated in brown, TFIIH subunits are indicated in grey, TRiC subunits are indicated in orange, the COP9 subunits in yellow, CRL subunits in dark purple and TC-NER factors are indicated in red. **C** IP of CSA-mClover and GFP-DDB2 using GFP beads in CSA-mC and GFP-DDB2 KI cells followed by immunoblotting for the indicated proteins. HCT116 cells were used as a control. The experiment was repeated two times with similar results. Source data are provided as a Source Data file.

analysis, UV colony survival and FRAP analysis (Supplementary Fig. 2B, C). Pulldown of GFP-DDB2 from these cells followed by both immune-blot and LC-MS analysis, (Fig. 1C, Supplementary Fig. 2D and supplementary Data 2) confirmed that DDA1 could not be detected in CRL4$^{DDB2}$ complexes. It should be noted, however, that the absence of DDA1 identification within CRL4$^{DDB2}$ does not fully prove its absence in this complex as it might associate sub-stoichiometric or only transiently.

## DDA1 is a component of the CSA/DDB1 complex

To investigate the binding of DDA1 to CSA, we assembled part of the TC-NER ubiquitin ligase complex and determined its structure by cryo-EM at 3.4 Å resolution (Fig. 2A–E, Supplementary Figs. 3–4 and supplementary Table 1). The complex contains CSA-DDB1-DDA1, the substrate recognition module of CRL4A$^{CSA}$, together with K414 mono-ubiquitinated UVSSA[14] and the catalytic inactive deubiquitinating enzyme USP7 (USP7$^{C223A}$). The first and third WD40 domains of DDB1 (BPA and BPC), CSA as well as the N-terminal region of DDA1 were well resolved, whereas the exposed second WD40 domain of DDB1 (BPB) has multiple orientations relative to the other domains of DDB1. After focused refinement, the VHS domain of UVSSA was also well defined, and its interaction with CSA was found similar to that in a recently published cryo-EM structure (PDB 7OO3)[21] (Fig. 2B–D). However, USP7 and the rest of UVSSA showed high heterogeneity and the USP7 structure could not be resolved even after various attempts at local processing.

In this structure we found DDA1 to interact with both DDB1 and CSA. The DDA1 interaction with DDB1 is similar to previous structures: the DDB1-DDA1 complex (PDB 6DSZ) and the RBM39-DCAF15-DDB1-DDA1 complex (PDB 6Q0W, 6UD7, 6PAI)[36–39], involving residues 2-75 of DDA1. The interaction of DDA1 with CSA is in a similar area of this DCAF as the DDA1 interaction with DCAF15, but the details are different, as DDA1 has clearly rearranged itself. In the DCAF15-DDA1 complex, DDA1 forms an α helix from residue 53 to 73, creating a binding interface that complements an extensive groove of DCAF15. The DDA1 binding groove of DCAF15 is not conserved in CSA (Supplementary Fig. 5A, B), and we did not observe the full helix in the interaction with CSA. Instead, low-resolution densities were identified around blades 4 and 5 of the CSA β propeller. We could visualize DDA1 up to residue 61, but the region where it binds to CSA was poorly resolved and we could not

build any model with confidence (Fig. 2D, E). By applying a mask and focused processing, we found that the densities in this area adopt multiple conformations, suggesting that the interaction is at low affinity and transient. To strengthen our observation, the conformations of DDA1 with CSA-DDB1 are predicted by AlphaFold2 multimer[40,41]. The predicted models are highly consistent with each other and predicted with high confidence. They correlate well with our cryo-EM structure, including the C-terminal helical part interacting with CSA (Supplementary Fig. 5C–E). We hypothesize that in order to be able to interact with various targets, DDA1 has adopted unique interactions with different DCAF subunits.

To better understand the role of DDA1 in the complex, we tested its effect on the thermal stability of the complex using nano-differential scanning fluorimetry. The addition of DDA1 conferred a modest but reproducible stabilization of one degree in melting curve analysis compared to CSA-DDB1 alone (Supplementary Fig. 5F, G). The complex with DDA1 C-terminal truncation (CSA-DDB1-DDA1$^{1-52}$), in which the CSA interacting helix is deleted, showed similar result stabilization as the full-length DDA1, indicating that the protection effect is mainly contributed by the part interacting with DDB1. This result is in agreement with our structural observation that DDA1 has limited interface with CSA and could explain why DDA1 has less thermal protection effect in contrast to the DCAF15 DDB1 complex[37]. Nevertheless, this result suggests that DDA1 plays a structural and possible stabilizing role in the CRL4$^{CSA}$ complex. Although this stabilizing effect is not directly through protecting CSA, it could protect the integrity of the whole E3 ligase and may be required to provide sufficient dynamics of the complex. Recent structural studies have elucidated how CRL4$^{CSA}$ is assembled within the TC-NER complex in a context with RNAPII and CSB[21,42]. To understand how DDA1 is arranged in this complex, we superimposed our structure into the PolII-ELOF1-TCR complex[42]. In the model, the C-terminal helix of DDA1 passes through a cavity surrounded by CSA, DDB1 and CSB (Supplementary Fig. 6A, B). The extension of the C-terminal helix is exposed near the ATPase domain of CSB.

To examine and/or confirm whether DDA1 would interact with CSA, DDB1 and CSB, we applied crosslink mass spectrometry[43] (XL-MS). We assembled the CSA-DDB1-DDA1 complex, together with UVSSA, USP7 and CSB. The complex was cross-linked and the bound proteins were digested into covalently cross-linked peptides.

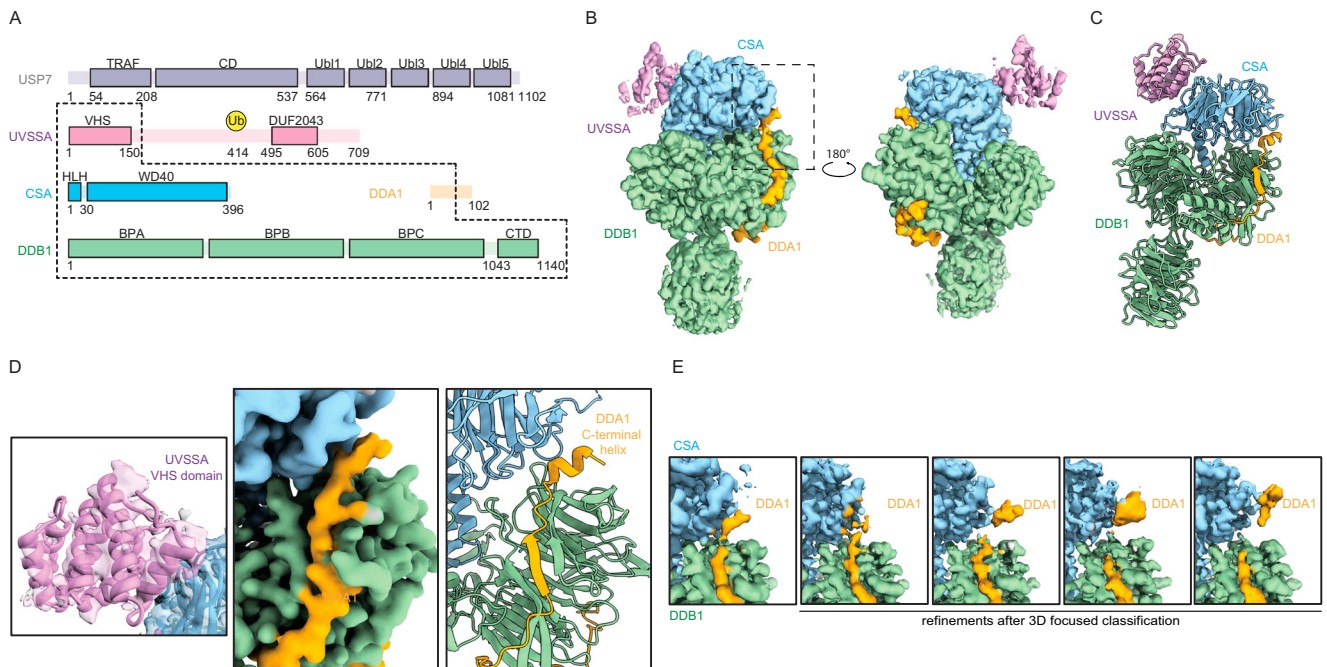

**Fig. 2 | DDA1 is a component of CRL4$^{CSA}$ complex. A** Domain architecture of the protein complex used for cryo-EM analysis. The stable core is highlighted by dash lines. Due to structural heterogeneity, USP7 and a large part of UVSSA are invisible in the final reconstructed cryo-EM map. **B** Cryo-EM structure of UVSSA-CSA-DDB1-DDA1. CSA (in light blue) and DDB1 (in light green) form a canonical substrate recognition module of CRL4 E3 ligases. The VHS domain of UVSSA (in pink) binds to a corner of CSA. DDA1 (in yellow) interacts with both DDB1 and CSA. **C** Molecular model of UVSSA-CSA-DDB1-DDA1 in ribbon diagram. **D** Close-up views of CSA interacting proteins. UVSSA interacts with CSA via the VHS domain (in pink). The C-terminal helix of DDA1 (in yellow) interacts with CSA. **E** Unstable interaction between DDA1 and CSA. The C-terminal helix of DDA1 is poorly resolved and various forms of densities can be identified by focused classification on this region, indicating that the interaction is unstable. Source data are provided as a Source Data file.

Identification of cross-linked peptides, by MS and analyzed using MaxQuant[44] software with integrated MaxLynx[45] revealed residues in close spatial proximity. We identified 140 high-confidence inter-protein cross-links in total (Supplementary Fig. 6C and supplementary Data 3). Importantly, we observed 30 inter-protein cross-links between DDA1 with CSA, DDB1, and CSB, involving DDA1 residues Lys13, Lys26, Lys51, Lys65, Lys66 and Lys70 (supplementary Data 3). Remarkably, no crosslink peptides were found between DDA1, UVSSA and USP7. Although this does not provide information about specific residues that mediate the interaction, the location of lysine 51, 65, 66 and 70 residues in DDA1 and their association with the outer regions of the β-propeller blades made up by the WD40 domain of CSA and C-terminal domain CSB are in line with the cryo-EM comparison in which DDA1 is located within a cavity created by CSA, DDB1, and CSB (Supplementary Fig. 6A, B). The XL-MS supports the cryo-EM data and strengthens the role of DDA1 as a component of the TC-NER complex.

## DDA1 is required for transcription recovery following DNA damage
CRL4$^{CSA}$ is a crucial TC-NER factor in resolving TBLs and the subsequent resumption of transcription arrested by DNA damage. To investigate the significance of DDA1 for the CRL4$^{CSA}$ function in TC-NER we tested the effect of its absence (DDA1KO, Fig. 3A and Supplementary Fig. 7A) on CSA and CSB stability, UV-survival and recovery of RRS after UV treatment, which is a proxy for TC-NER activity[28]. The levels of CSA and CSB protein were unaltered in total extracts isolated from DDA1KO cells compared to the parental cell line (Fig. 3A), revealing that DDA1 loss does not affect the stability of CSA and CSB. DDA1-KO cells showed a clear RRS defect (Fig. 3B) after 10 J m$^{-2}$ UV irradiation, similar to CSA and CSB KO cells. However, in contrast to this strong RRS defect, DDA1KO cells were only moderately sensitive to UV irradiation when compared to CSA and CSB KO cells, as measured by

colony survival assay (Fig. 3C). Strikingly, at lower UV doses, the DDA1-dependent UV sensitivity was not even significantly distinct from WT cells and became only apparent at higher doses. We speculated that this moderate sensitivity of DDA1-KO cells at low doses of UV would correlate to a similar mild effect on RNA synthesis resumption at lower UV doses. Indeed, RRS did not appear to be affected at lower doses of 2.5 and at 5 J/m$^2$ UV in DDA1KO cells and became only noticeable at a higher dose of 10 J m$^{-2}$ (Fig. 3D). These data suggest that the crucial role of DDA1 in TC-NER becomes apparent at higher UV doses, when more functional CSA-containing complexes are likely required. Importantly, the ectopic expression of DDA1 in DDA1KO cells did rescue the resumption of transcription (Supplementary Fig. 7B), indicating that the observed phenotypes are directly associated with the DDA1 protein. Additionally, we investigated whether the C-terminal region of DDA1, located within the cavity created by CSA, DDB1, and CSB, would affect RNA recovery (Supplementary Fig. 6A, B). The transcription resumption was impaired in DDA1KO cells ectopically expressing the truncated version of DDA1$^{1-52}$ (DDA1Δ) and in DDA1$^{1-65}$ HCT116 cells generated by CRISPR-Cas9 (Supplementary Fig. 7C–E). The data confirm the important role of DDA1 and highlight the function of its C-terminal region in TC-NER (Supplementary Fig. 7F, G).

We demonstrated that in DDA1KO cells, RNA synthesis in absence of DNA damage, gene expression and global protein levels are not affected, and are overall comparable to WT cells (Supplementary Fig. 8A–F and supplementary Data 4–6), suggesting a direct role of DDA1 in TC-NER. However, since DDA1 is part of multiple CUL4-based E3 ligases, we cannot exclude that the observed phenotype may be partially due to a pleiotropic effect.

## DDA1 is required for proper CSA localization
Previously, we found that the TRiC chaperonin is required for proper folding, stability and incorporation of CSA into the CRL4 complex and

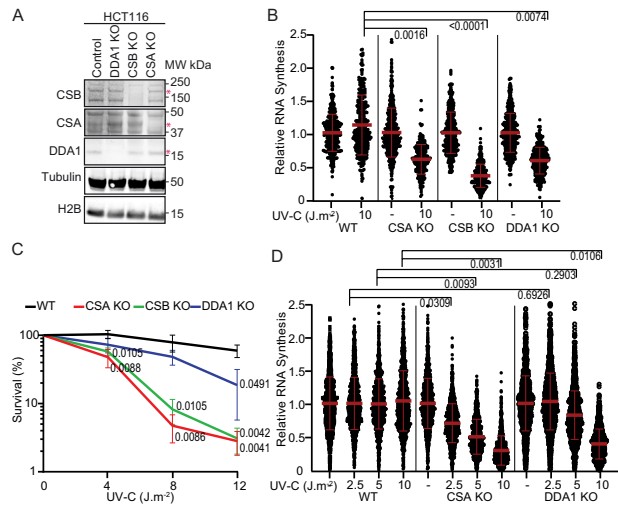

**Fig. 3 | DDA1 is required for transcription recovery following DNA damage.**
**A** Immunoblot of cell extracts from the HCT116 WT and KO cells was stained for the indicated proteins. Tubulin and H2B were used as loading control. The experiment was repeated two times with similar results. **B** Transcription restarts after UV, determined by relative EU incorporation in HCT116 WT and KO cells, at 24 hours after UV exposure (10 J m⁻²). EU incorporation-derived fluorescence was normalized to non-irradiated cells (set to 1). The mean ± S.D. is indicated in red from three independent experiments of (left to right) $n = 454, 370, 506, 231, 393, 348, 460,$ and $297$ cells. **C** UV colony survival of HCT116 WT and KO cells exposed to the indicated doses of UV. Graphs depict the mean ± SD from three independent experiments, the numbers represent $p$ values. $p$ values ≤ 0.05 were considered significant relative to WT analyzed by unpaired, two-tailed $t$ test, adjusted for multiple comparisons. **D** Transcription restarts after UV, determined by relative EU incorporation in HCT116 WT and KO cells, at 24 hours after UV exposure (2.5, 5, and 10 J m⁻²) or mock-treated. EU incorporation-derived fluorescence was normalized to non-irradiated cells (set to 1). The mean ± S.D. is indicated in red from three independent experiments of (left to right) $n = 1174, 1272, 1219, 1168, 1275, 1235, 1148, 980,$ $1014, 1278, 1166,$ and $1039$ cells. Data shown in **B** and **D** numbers represent $p$ values (nested $t$ test, two-tailed). Source data are provided as a Source Data file.

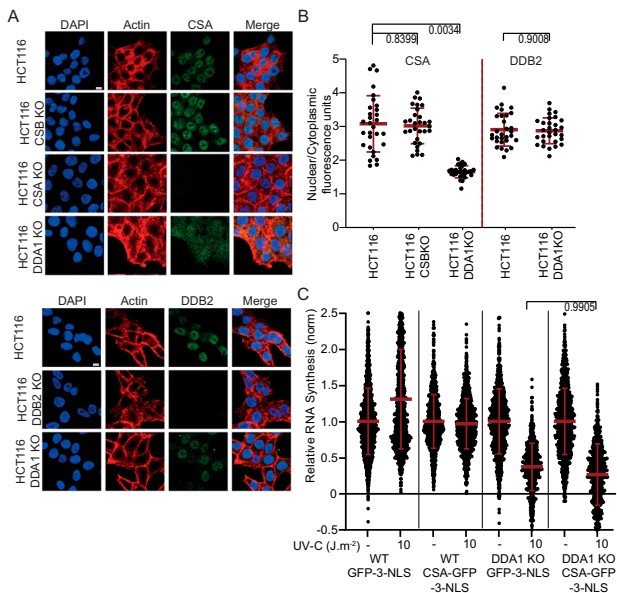

**Fig. 4 | DDA1 provides properly CSA localization.** **A** Representative immunofluorescence images of endogenous CSA and DDB2 in HCT116 WT and KO cells, scale bar: 10 μm. **B** Nuclear and cytoplasmic CSA and DDB2 levels in HCT116 WT and KO cells, analyzed and quantified by fluorescence microscopy and ImageJ. The mean ± S.D. is indicated in red from three independent experiments of (left to right) $n = 30, 30, 30, 30,$ and $30$ images. CSA and DDB2 signal intensity at the nucleus (as identified by DAPI staining) was compared to that in the rest of the cell (phalloidin). **C** Transcription restarts after UV damage as determined by relative EU incorporation in HCT116 WT and KO cells, with either CSA-GFP-3NLS or GFP-3NLS expression, 24 h after UV exposure (10 J m⁻²) or mock-treated. EU incorporation levels were normalized to the non-irradiated cells (set to 1). The mean ± S.D. is indicated in red from three independent experiments of (left to right) $n = 1275, 1118, 1164, 1195,$ $1068, 603, 1061,$ and $677$ cells. Data shown in **B**, **C** numbers represent $p$ values (nested $t$ test, two-tailed). Source data are provided as a Source Data file.

its subsequent nuclear localization, essential for optimal performance of the CRL4$^{CSA}$ complex in TC-NER[33]. In contrast to TRiC, DDA1 does not seem to be necessary for CSA's stability (Fig. 3A). However, immunofluorescent analysis of endogenous CSA showed that in the absence of DDA1, CSA is not exclusively localized to the nucleus and shows increased levels in the cytoplasm as compared to wild-type and CSB KO cells, assayed in parallel (Fig. 4A, B). This observation was confirmed by siRNA against DDA1 in cells ectopically expressing GFP-tagged CSA, showing that loss of DDA1 triggers mislocalization of CSA (Supplementary Fig. 9A, B). Moreover, expressing GFP-tagged DDA1 in the DDA1-KO cells not only restored the resumption of transcription after UV-induced inhibition (Supplementary Fig. 7B) but also rescued the subcellular localization of CSA (Supplementary Fig. 9D). Interestingly, absence or depletion of DDA1 did not affect nuclear localization of endogenous DDB2 nor ectopically expressed GFP-tagged DDB2 (Fig. 4A, B and Supplementary Fig. 9C). Together the data showed that DDA1 specifically modifies CRL4$^{CSA}$ without affecting the similar GG-NER-specific CRL4$^{DDB2}$ complex.

This partial CSA nuclear localization in DDA1-deficient cells may be causative for the observed TC-NER defect. It is thus expected that, by restoring the correct cellular compartmentalization of CSA, the repair capacity could be complemented. Strikingly however, expression of GFP-CSA fused to an array of three nuclear localization signals (3NLS), CSA-GFP-3NLS (Supplementary Fig. 9E) in DDA1KO cells did not rescue the transcription resumption in response to UV irradiation (Fig. 4C), despite that the addition of 3NLS to CSA provided full nuclear localization. These experiments strongly suggest that the TC-NER

defect in DDA1KO cells is associated with another, thus far unidentified, molecular mechanism rather than a reduced nuclear protein level of CSA.

## DDA1 modulates the protein network of CRL4$^{CSA}$ complex

To further assess how DDA1 is linked to CRL4$^{CSA}$ function, we examined whether the loss of DDA1 would affect the protein network of CSA. To that aim, we immunoprecipitated CSA-mClover from WT KI and DDA1KO (CSA-mClover KI) cell lines after UV irradiation followed by MS applying the data independent acquisition[46] (DIA) and label-free quantification (LFQ) approach[47]. DIA is a recent MS approach which provides high sensitivity with unprecedented proteome coverage[48] (supplementary Data 7). MS comparison between CSA-mClover and non-expressing control cells (Supplementary Fig. 10A, B) recapitulated our previous MS results (Fig. 1), substantiating these and our earlier observations[33]. We observed a stronger interaction of CSA with RNAPII, PAF-complex, CSB, UVSSA and the CRL and COP9 complexes after UV-induced DNA damage in cells lacking DDA1 (Fig. 5A). This increased interaction might be caused by the reduced TC-NER activity in the absence of DDA1 in which factors still assemble but remain associated. However, even without DNA damage and despite part of CSA roaming in the cytoplasm, we noted a stronger interaction of CSA with CSB, UVSSA and the core subunits of the CRL and the COP9 complexes (Fig. 5B), suggesting that DDA1 is affecting the protein network of the CRL4$^{CSA}$ complex already under basal conditions. The interaction of CSA with TRiC complex was unchanged and not affected by UV or loss of DDA1 (Supplementary Fig. 10C). The MS data were confirmed by immunoprecipitation and immunoblot analysis (Supplementary Fig. 11A–C) These data suggest that in the absence of DDA1 there is

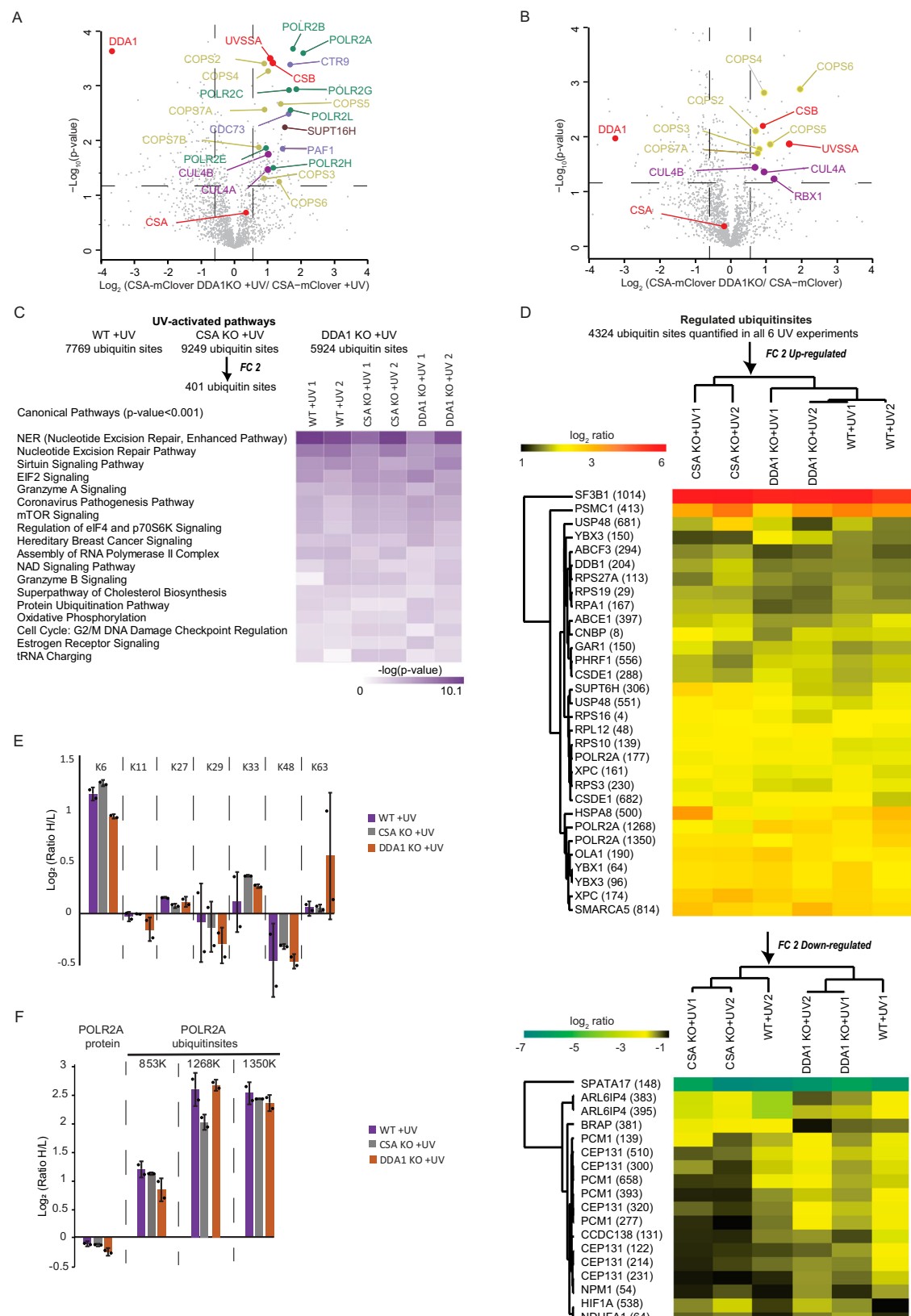

either an imbalance in complex assembly, or, due to the diminished nuclear presence of CSA, a larger proportion of CRL4$^{CSA}$ is engaged in TC-NER in response to endogenously produced TBLs, or, alternatively, DDA1 is involved in the dynamic turnover of the CRL-COP9 complex and in partial pre-assembled TC-NER complex.

## CSA- and DDA1-dependent ubiquitination

The cellular response to UV irradiation triggers a series of ubiquitination events that facilitate TC-NER progression in which the CRL4$^{CSA}$ complex[14,15] plays an important role together with other ubiquitin ligases[49–51]. To investigate whether the impaired TC-NER performance

**Fig. 5 | DDA1 modulates the protein network of CRL4$^{CSA}$ complex. A, B** Volcano plots depicting the statistical differences between three replicates of the MS analysis after GFP immunoprecipitation of UV-treated (**A**) or mock-treated (**B**) cells, comparing the protein network of CSA in WT with DDA1KO cells. The fold change (Log$_2$) is plotted on the *x* axis and the significance (*t* test −Log$_{10}$ (*P* value), Two-sample test, two-tailed) is plotted on the *y* axis. RNAPII subunits are indicated in green, PAF-1 subunits are indicated in light purple, proteins associated with RNAPII are indicated in brown, the COP9 subunits in yellow, CRL subunits in dark purple and TC-NER factors are indicated in red. **C** Heatmap showing the statistically significantly enriched canonical pathways (*p* value 0.001, Ingenuity Pathway Analysis, IPA) of the UV-responsive ubiquitin sites that passed a twofold change cutoff (including duplicates). The color coding depicts −Log$_{10}$(*P* value) (Fisher's Exact Test) of the statistically significant terms. **D** Heatmap showing the Log$_2$ SILAC ratios of ubiquitin sites that are quantified in all UV conditions (including duplicates) over untreated controls and that passed a twofold change cutoff (up and down-regulated). The color density reflects the scale of enrichment. **E** Log$_2$ SILAC ratios of ubiquitin K6, K11, K27, K29, K33, K48, and K63 chains as determined by quantitative global ubiquitin-proteomics in WT, CSAKO, and DDA1KO cells after UV treatment (20 J m$^{-2}$, 30 min). The mean ± S.D. of duplicate experiments are plotted. **F** Log$_2$ SILAC ratios of POLR2A protein and ubiquitin sites of POLR2A (853 K, 1268 K, and 1350 K) as determined by quantitative proteomics and global ubiquitin-proteomics in WT, CSAKO, and DDA1KO cells after UV treatment (20 J m$^{-2}$, 30 min). The mean ± S.D. of duplicate experiments are plotted. Source data are provided as a Source Data file.

in DDA1KO cells is related to altered CRL4$^{CSA}$ ubiquitination activity by its increased association with the inhibitory COP9 subcomplex, we conducted global UV-induced ubiquitin signaling to profiling by SILAC-based MS in WT, CSAKO and DDA1KO cell lines. Ubiquitinated peptides were enriched by immunoaffinity purification using an antibody bound to a resin that specifically recognizes diglycine (diGly)-modified peptides (K-GG)[52], generated by tryptic digestion of ubiquitin-modified proteins. We performed, in duplicate, three separate experiments in each of which Light- and Heavy-SILAC cells were mock or UV treated (Supplementary Fig. 12A). In addition, we also compared WT cells with either CSA or DDA1KO cells without external DNA damage induction to identify possible basal or intrinsic CSA-, and DDA1-dependent ubiquitination activity. Together, this led to the identification of 23,054 unique ubiquitin sites, which were reduced to 17,697 unique sites after stringent filtering (Supplementary Fig. 12A, supplementary Data 8). The ubiquitinome coverage distribution plot showed similar quantification depths for all conditions and good reproducibility between experimental duplicates (Supplementary Fig. 12B, C, Supplementary Fig. 13A–E, Supplementary Fig. 14A–C).

To identify UV-activated pathways, we focused on proteins containing ubiquitin sites found in both duplicates in at least one of the tested cell lines after UV irradiation. The application of a threshold filter of two fold change provided us 401 UV-induced ubiquitin sites, which were subjected to IPA[53]. As expected, UV irradiation-induced strong activation of DDR manifested by the significant enrichment of NER pathway components among these sites (Fig. 5C). Additionally, K6-linked ubiquitin chains were substantially increased, in contrast, the levels of K48 and K63 were almost unaltered after UV (Fig. 5E), in line with previous observations[54,55], which validated the applied procedure and obtained results. Strikingly, no prominent overall changes were detected in UV-activated pathways between WT, CSAKO and DDA1KO cell lines.

For an in-depth comparison of the ubiquitination responses upon UV irradiation among WT, CSAKO, and DDA1KO cell lines, we examined ubiquitin sites which were quantified in all six UV experiments (4324 unique ubiquitin sites) and we applied a threshold filter of two fold change, resulting in 49 ubiquitin sites found to be commonly responding upon UV irradiation (Fig. 5D). We also observed specific ubiquitin-peptides in CSB and UVSSA, which were previously identified as important UV-induced ubiquitination targets[15,16], however, they were not consistent among the experiments (i.e., K414 in UVSSA, Supplementary Fig. 14A–C, and supplementary Data 8), This inconsistency makes it challenging to draw definitive conclusions. However, it is possible that the CSB and UVSSA ubiquitination is a highly dynamic phenomenon occurring in a specific or short time window during TC-NER, which may not be easily captured by our procedure at the set time after UV irradiation. To further investigate the role of DDA1, we performed in vitro ubiquitination assays with CSB, UVSSA and neddylation-activated CRL4$^{CSA}$ complex with and without DDA1 (Supplementary Fig. 15A–C). Under those conditions, we observed robust and fast polyubiquitination of CSB and mono-ubiquitination of UVSSA, showing that, indeed, CSA is capable of ubiquitinating CSB and UVSSA.

These data further suggest that DDA1 is not essential for the E3 ligase CRL4$^{CSA}$ activity on CSB and UVSSA in vitro. However, in cells, the CRL4$^{CSA}$ ubiquitination activity is negatively regulated by the COP9 signalosome through its capacity to remove the CRL-activating NEDD8 from CUL4[13,56]. Since we noted that in non-treated conditions (Fig. 5B) CSA is strongly associated with CSB, UVSSA, CRL and the COP9 complex in DDA1KO cells, it might be expected that the ubiquitin profile is already altered in the absence of exogenous DNA damage. Indeed, ubiquitin profiling of DDA1KO cells without exogenous DNA damage showed that a strikingly large number of ubiquitin sites and pathways were differentially regulated in the absence of DDA1 (Fig. 6A), endorsing our hypothesis. Among the pathways that were affected without exposure to exogenous genotoxic agents, also several DDR-associated processes, including NER, were identified. This DDA1-dependent alteration of the ubiquitin signaling profile, suggests that DDA1, potentially through its role in CRL4$^{CSA}$, is required to maintain cellular homeostasis even without exposure to exogenous DNA damaging agents. Similar significant pathway changes were observed in non-irradiated CSAKO cells, corroborating that DDA1-function within CRL4$^{CSA}$ is already effective under basal conditions, *i.e.* without exposure to exogenous DNA damaging agents.

Our findings indicate that under non-damaging conditions, several pathways were affected in the absence of either DDA1 or CSA, ranging from DDR-associated signalling, mRNA processing, translational mechanism, mitochondrial function and protein folding/stability processes. Strikingly, with previous gene expression profiling analysis similar biological pathways were found to be altered during aging, in NER-deficient mouse cells and upon low doses of UV exposure[57]. Further in-depth analysis has provided evidence that these altered gene expression profiles were mainly caused by transcription stress, induced by endogenous DNA damage[58]. It is thus conceivable that CRL4$^{CSA}$, including DDA1, is activated by the presence of endogenous DNA damage. Additionally, we conducted an in-depth comparison of the ubiquitination profiles in the absence of DNA-damaging agents. We specifically examined ubiquitin sites that were quantified in all mock experiments (2947 unique ubiquitin sites). Unfortunately, several ubiquitin sites were found to be inconsistent among all experiments and were therefore not considered in the comparison. We applied a threshold filter of two fold change, resulting in the identification of 41 ubiquitin sites that were downregulated in CSAKO cells. Interestingly, 10 of these ubiquitin sites were shared with DDA1KO, and even 18 (45%) when the cutoff was set at −0.5 Log$_2$. (Supplementary Fig. 15D and supplementary Data 9) Notably, this includes the K1268 ubiquitin site of RNAPII. Future experiments are required to reveal the significance of these ubiquitin sites.

## Differential ubiquitination of RNA polymerase II

We also observed specific ubiquitination of RNAPII's subunit RPB1/POLR2A (K177, K853, K1268, and K1350) after UV irradiation (Fig. 5F), in line with earlier studies[14,20]. Unexpectedly, we found that the SILAC ratio of UV-induced K1268 ubiquitination of RNAPII is only slightly decreased in the full absence of CSA. This specific ubiquitination site

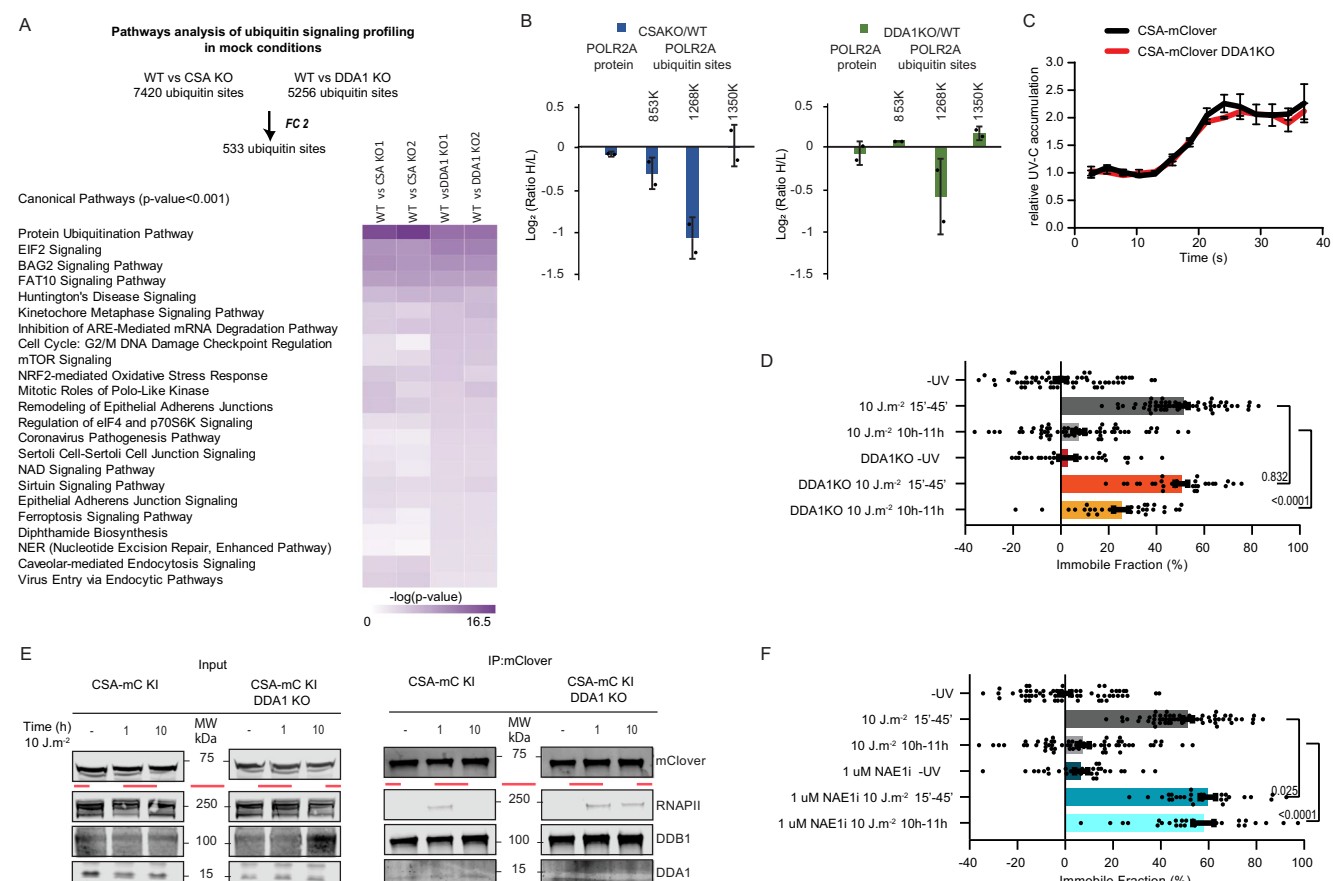

**Fig. 6 | DDA1 affects the dynamic of CRL4^CSA via COP9 complex. A** Heatmap showing the statistically significantly enriched canonical pathways (*p* value 0.001, Ingenuity Pathway Analysis, IPA) of the ubiquitin sites, which were differentially modulated in mock-treated conditions, that passed a twofold change cutoff. The color coding depicts −Log₁₀ (*P* value) (Fisher's Exact Test) of the statistically significant terms. **B** SILAC ratios of POLR2A protein and ubiquitin sites of POLR2A (853 K, 1268 K, and 1350 K) as determined by quantitative proteomics and global ubiquitin-proteomics in WT/CSAKO and WT/DDA1KO cells in mock-treated conditions. The mean ± S.D. of duplicate experiments are plotted. **C** Binding kinetics of CSA-mClover in HCT116 WT or DDA1KO cells to locally UV damaged sites induced by 266 nm micro-beam laser irradiation. GFP fluorescence intensities at the site of UV damage were measured by real-time imaging until they reached a maximum. Graphs depict the mean & S.E.M. of *n* = 30 cells per condition from three independent experiments. **D** FRAP analysis of CSA-mClover in mock or UV irradiated (10 J m⁻²) HCT116 WT and DDA1KO cells, measured at the indicated time points. The

percentage of CSA-mClover immobile fraction was determined from FRAP analyses (Supplementary Fig. 16B). Graphs depict the mean & S.E.M. of (top to bottom) *n* = 60, 60, 60, 30, 30, and 30 cells from at least three independent experiments. **E** IP of CSA using GFP beads in CSA-mC KI HCT116 WT and DDA1KO cells followed by immunoblotting for the indicated proteins. Cells were collected 1 and 10 h after mock-treatment or irradiation with (10 J m⁻²) UV. The experiment was repeated two times with similar results. **F** FRAP analysis of CSA-mClover in the presence or absence of NAEi added 0.5 h before irradiation and followed by UV irradiation (10 J m⁻²). The percentage of CSA-mClover immobile fraction determined from FRAP analyses (Supplementary Fig. 16D) was measured at the indicated time points. Graphs depict the mean & S.E.M. of (top to bottom) *n* = 60, 60, 60, 30, 30, and 30 cells from at least three independent experiments. Data shown in **D** and **F** numbers represent *p* values (unpaired, two-tailed *t* test adjusted for multiple comparisons). Source data are provided as a Source Data file.

was previously identified as a focal point for coordinating TC-NER, protein assembly and signalling RNAPII for degradation and essential for surviving genotoxic insults[14,59,60]. Since we observed multiple CSA/DDA1-dependent ubiquitination events in the absence of exogenous DNA damage (Fig. 6A and discussed above), we further specifically focused on the RPB1 ubiquitination sites. Strikingly, we observed a clearly reduced RPB1-K1268 ubiquitination in CSAKO and a less prominent reduction in DDA1KO cells compared to isogenic WT control cells under non-damaging conditions. On the other hand, other known UV-responsive ubiquitin sites on RPB1, K853 and K1350 were not affected (Fig. 6B) by loss of CSA or DDA1 in the absence of exogenous DNA damage. Importantly, the protein level of RNAPII was not changed, as deduced from the global protein profile determined by MS using the same input material applied for the detection of diGly-modified peptides (Fig. 5F, supplementary Data 6). Hence, it appears that CRL4^CSA constitutively ubiquitinates this lysine residue 1268 in RPB1, a process crucial for the progression of TC-NER, which seems to

be modulated by DDA1. The precise biological role of this constitutive ubiquitination at RPB1-K1268 remains elusive, especially considering that mice carrying this mutation do not exhibit evident phenotypic changes in the absence of externally induced TBLs[14]. We can only speculate on the trigger for this CRL4^CSA-mediated activity, which may arise from the persistent albeit low levels of TBLs by endogenous sources of DNA damage[61–63] (i.e., cellular metabolites such as ROS or aldehydes).

It should be noted, however, that the UV-induced ubiquitin profiling reveals relative differences in RPB1-K1268 ubiquitination rather than absolute values. We thus speculate that the previously proposed essential role of CSA to ubiquitination RPB1-K1268 in response to UV-irradiation is, in part, explained by the intrinsic reduced K1268 ubiquitination in the absence of exogenous DNA damage when compared to WT cell (Fig. 6B). Although the majority of RPB1-K1268 ubiquitination is CSA dependent, it is also likely that a part of the TBL-induced RPB1 ubiquitination may occur by a CSA independent event.

## DDA1 controls CRL4$^{CSA}$ activity via the COP9 complex

CRL4$^{CSA}$ activity appears important for the basal level of RNAPII ubiquitination, thereby also significantly reducing the overall ubiquitination response to UV-induced TBLs. However, Vidaković et al.[20], suggested that the CSA-dependent RPB1-K1268 ubiquitination became mainly apparent at later time points post UV. In this scenario, the role of DDA1 as modulator of CSA, would become more evident in the TC-NER process at later time points. Since we observed that the absence of DDA1 partly impairs the RPB1-K1268 ubiquitination, though not to the same extent as the absence of CSA, we investigated whether the loss of DDA1 (Supplementary Fig. 16A) would affect the dynamic association of CSA-mClover with the TC-NER machinery by live cell imaging. CSA rapidly accumulated at the site of UV laser-induced DNA damage and was not influenced by the absence or presence of DDA1 (Fig. 6C), indicating that TC-NER complex assembly is not affected. FRAP analysis of CSA-mClover, shortly after UV-irradiation also showed that immobilization of CSA, reflecting binding to lesion-stalled RNAPII, was not changed by the absence of DDA1. However, FRAP analysis at a later time point, i.e., 10 h post UV, showed that a significant fraction of CSA-mClover molecules remained immobilized in DDA1KO cells, whereas in WT cells, the mobility was fully recovered to the same level as in undamaged cells (Fig. 6D, Supplementary Fig. 16B). Similar results were obtained using CSB-mClover KI cell line[32] (Supplementary Fig. 16C). These data suggest that in the absence of DDA1, CSA and CSB molecules were longer bound to lesion-stalled RNAPII. This was confirmed by immunoprecipitation of CSA-mClover and immunoblot analysis (Fig. 6E), showing that the interaction between CSA and RNAPII was still evident 10 h after UV-irradiation in the absence of DDA1, whereas in WT cells, this was not observed. This reduced clearance of CSA-mClover from lesion-stalled RNAPII can either be caused by a reduced repair rate (longer presence of TBLs) or a slower disassembly of regulatory subunits from the CRL4$^{CSA}$ complex. The latter option is in line with the increased interaction between CRL4$^{CSA}$ with the COP9 complex in the absence of DDA1 in UV irradiated, but also in mock treated as detected by MS (Fig. 5A, B). These data suggest an intrinsically slower disassembly of the CRL4$^{CSA}$-COP9 complex when DDA1 is absent. We speculate that this aberrant complex turnover might be related to a compromised CRL4$^{CSA}$ activity in which the presence of COP9 complex may physically interfere with its activity.

CRL is activated by NAE1, which conjugates the ubiquitin-like NEDD8 to CUL4. The covalent attachment of NEDD8 induces a conformational change of CUL4, thereby promoting polyubiquitination of its substrates. However, the CRL-associated COP9 complex keeps CRL4$^{CSA}$ in a dormant state by removing NEDD8 from CUL4. Regulation of CRL activity is thus achieved by a delicate balance between activating and de-activating modalities. Disturbance of this balance or increased association with COP9 complex, as in the absence of DDA1, may thus interfere with a timely activation/inactivation cycle of the complex. To gain insight into the molecular events that mediate CRL4$^{CSA}$ activity and the TC-NER progression, we inactivated this complex with the broad class neddylation inhibitor, MLN4924[64] (NAEi). NAEi treatment triggers the accumulation of the demethylated, inactive isoform of the complex, mimicking its partial inactivation by increased COP9 association caused by DDA1 absence. The inhibitor treatment completely prevented CSA mobility from returning to the same level as in non-damaged cells 10 h after UV irradiation, similarly, though stronger, as observed in DDA1KO cells (Fig. 6F, Supplementary Fig. 16D). Altogether, these results further support the idea that DDA1 is important for the molecular coordination and dynamics of the CRL4$^{CSA}$ by tuning the COP9 complex interaction.

## Discussion

The mechanism how CRL4$^{CSA}$ drives TBLs removal and its significance for transcription maintenance are still unclear. Although the CRL4$^{CSA}$ E3 ligase structure is well defined, the identification and characterization of its interacting DDA1 factor revealed a more complex organization, in which DDA1 plays an important role in promoting resolution of TBLs, by modulating the CRL4$^{CSA}$ ubiquitinating activity to control transcription-coupled repair. Our study reveals an unexpected complexity of how regulatory ubiquitination orchestrates the progress of TC-NER.

CRL4s are a large family of E3 ligases in which DDB1-CUL4 associated factors (DCAF) are receptors to identify a great number of specific substrate proteins[65]. DDA1 is a core subunit of multiple, but not all, CRL4 complexes. Remarkably, DDA1 was not found to be associated with or dynamic bound to the CRL4$^{DDB2}$ and its absence did not influence DDB2's nuclear localization nor its function in GG-NER. This finding is intriguing, since the overall architectures of both CRL4$^{DDB2}$ and CRL4$^{CSA}$ E3 ubiquitin ligase complexes appear very similar[11], although the major differences in enzymatic activity are defined by the WD40 domain of each DCAF. This domain coordinates the CRL4-E3 ligase activity by functioning as an interaction platform for the binding of specific proteins to diversify the substrate range. Our findings suggest that each DCAF has to adopt a strategy to deal with DDA1, providing an opportunity for a more subtle regulation of the CRLs. Indeed, we have found that the adding of DDA1 to CSA/DDB1 complexes in vitro confers a modest stabilization, in contrast to a more stabilizing effect for DCAF15/DDB1 where a DDA1-binding groove is present[36–39]. The stability observed was linked to the N-terminal region of DDA1, which is associated with DDB1. Intriguingly, the C-terminal helix of DDA1 was found to reside within a cavity formed by CSA, DDB1, and CSB. Supported by XL-MS data (Supplementary Fig. 6C), this suggests that the C-terminal region of DDA1 could potentially interact with other TC-NER components, such as CSB. The biological significance of this discovery was underscored by RRS experiments. However, the precise mechanisms through which DDA1 influences the overall topology of the fully assembled CRL4–substrate complex, along with its protein partners, remain to be elucidated.

CSA is regulated by several factors, including the chaperonin TRiC[33], which provides properly folded CSA to DDB1. We previously showed that the TRiC complex is required for CSA stability and to facilitate its assembly into the CRL4$^{CSA}$ complex, which can then efficiently translocate into the nucleus. Mutant CSA proteins, with likely exposed hydrophobic patches, lead to enhanced interaction with TRiC and cause cytoplasmic retention of CSA. Although we found that DDA1 promotes nuclear localization of CSA, quantitative MS and immunoblot revealed that DDA1 did not affect the level of CSA protein nor the interaction between CSA and TRiC, suggesting that the cellular localization of CSA in DDA1KO cells is not connected with the CSA folding, stability and with the hand-over mechanism for the formation of CRL4$^{CSA}$ complex. Based on these observations, it is tempting to speculate that loss of DDA1 may cause insufficient nuclear presence of CSA to fully support TC-NER at high loads of TBLs, even though the majority of the CRL4$^{CSA}$ complex is still nuclear. Most notable in this regard is that a tagged version of CSA fused with a tandem of three nuclear localization signals failed to rescue the TC-NER defect in DDA1KO cells. These observations suggest that the TC-NER-deficient phenotype is not only caused by a reduced nuclear protein level of CSA.

Importantly, we provide direct evidence that DDA1 controls the disassembly or dissolution of the TC-NER complex. Indeed, the mobility of CSA and CSB was impaired in DDA1KO cells, which correlates with a stronger interaction between CRL4$^{CSA}$ and the COP9 complex, and subsequently association with chromatin-bound RNAPII. Our findings suggest that the loss of DDA1 affects the dynamic activation-deactivation cycle of CRL4$^{CSA}$ through controlling the association of the COP9 complex. The presence of the DDA1 within the cavity, formed by CSA, DDB1, and CSB, may play a crucial role in finely tuning the complex. DDA1 may regulate the conformation of the RNAPII/TC-NER complex, influence its activity, and the directionality of the ubiquitin ligase activity of the CRL4$^{CSA}$ complex. By serving as a

ubiquitination controlling factor, DDA1 subsequently promotes the progress of the repair of DNA-damage-stalled RNAPII. We envisage a scenario in which the regulatory ubiquitination of RNAPII at TBLs is not only maintained by CRL4$^{CSA}$, but also involves other E3 ligases. The function of CRL4$^{CSA}$ may not be only to cooperate with other E3 ligases to trigger ubiquitination of RNAPII, but also to amplify, propagate and stabilize the initial ubiquitination events to coordinate the TC-NER process. The strongly altered ubiquitin profile in both CSA and DDA1KOs in the absence of external DNA damage may be explained by the continuous induction of TBLs from endogenous sources that may trigger the activation of transcription-coupled repair processes. In conclusion, our findings reveal that DDA1 is an important factor in regulating the progression of DNA repair, and it will be very interesting to gain structural insights into TC-NER, including DDA1 as a core component of CRL4$^{CSA}$ complex.

## Methods

### Cell lines and cell culture

HCT116 colorectal cancer cells were cultured in Dulbecco's modified Eagle's medium DMEM or in phenol red-free DMEM for a live cell imaging experiment, supplemented with 10% fetal calf serum, 1% penicillin/streptomycin in a humidified incubator at 37 °C and 5% CO2. VH10 (GFP-DDB2[66], hTert), CS3BE (CSA-GFP[33], hTert) fibroblasts were maintained in DMEM with 15% FCS and antibiotics. For SILAC, cells were grown for 2 weeks (>10 cell doublings) in arginine/lysine-free SILAC DMEM supplemented with 15% dialyzed FCS, 1% penicillin–streptomycin, 200 µg ml$^{-1}$ proline and either 73 µg ml$^{-1}$ light [$^{12}C_6$]-lysine and 42 µg ml$^{-1}$ [$^{12}C_6$, $^{14}N_4$]-arginine or heavy [$^{13}C_6$]-lysine and [$^{13}C_6$, $^{15}N_4$]-arginine.

HCT116 KO cells were generated by transiently transfecting HCT116 cells by jetPEI with a pLentiCRISPR.v2 plasmid[67] expressing Cas9 and containing appropriate sgRNAs (supplementary Data 11), according to manufacturer instructions. Transfected cells were selected by culturing in 1 µg ml$^{-1}$ puromycin-containing medium for 2 days, and single cells were seeded to allow expansion. Genotyping of single-cell clones was performed by immunoblotting or genomic PCR as indicated (supplementary Data 11).

CSA-GFP-3NLS, GFP-3NLS and GFP-DDA1 complemented cell lines were generated by lentiviral transduction in WT and DDA1$^{-/-}$ cells. Full-length expression construct with GFP-DDA1 and GFP-DDA1 Δ were synthesized (gene synthesis services, GenScript). Three nuclear localization signals (3NLS) were added to CSA-GFP and GFP[28]. Tagged CSA-GFP-3NLS, GFP-3NLS and GFP-DDA1 constructs were inserted in a pLenti-CMV-puro-DEST plasmid[68]. After transduction, cells were selected with 1 µg ml$^{-1}$ puromycin.

HCT116 CSA-mC KI cells were generated by transiently transfecting cells with a sgRNA-containing pLentiCRISPR.v2 plasmid (supplementary Data 11) targeting the stop codon of CSA and co-transfecting a homology-directed repair template, which included two TEV cleaving sites, mClover and on either side 300 bp CSA locus-specific genomic DNA for homologous recombination to each end of CRISPR-generated dsDNA break (gene synthesis services GenScript, sequence upon request). The cells were seeded and kept in the presence of 2 µg ml$^{-1}$ puromycin and subsequently sorted for mClover-positive cells by FACS. Single-cell clones were genotyped, and homozygous KI clones were selected for further analysis. Genotyping PCR was performed on genomic DNA isolated using a PureLink® Genomic DNA Mini Kit according to the manufacturer's protocol with Q5 hifi DNA polymerases according to the manufacturer's protocol. Primer sequences are provided in supplementary Data 11.

siRNA[69] (supplementary Data 11) transfections were performed 2 or 3 days before each experiment using Lipofectamine RNAiMax according to the manufacturer's protocol. Knockdown efficiency was determined by immunoblotting.

### Survival assays

For the clonogenic survival assay, 750 cells were seeded per well in triplicate in a sixwell plate. The following day, cells were treated with UV at the indicated doses. Following treatment, colonies were grown for 7–10 days, after which they were fixed and stained using Coomassie blue (50% methanol, 7% acetic acid and 0.1% Coomassie blue. Colony numbers were counted using GelCount. The relative colony number was plotted from at least three independent experiments, each performed in triplicate. Levels were normalized to mock-treated, set to 100 and plotted with standard deviation values (SD).

### RNA synthesis recovery assay

Cells were grown on coverslips and mock-treated or irradiated with 10 J m$^{-2}$ UV. RNA was labeled at the indicated time points for 1 h with 200 µM EU, fixed with 3.7% formaldehyde (FA) in PBS for 15 min at room temperature and permeabilized by 0.1% Triton X-100 in PBS for 10 min. Cells were incubated for Click-it-chemistry-based azide coupling for 1 h with 60 µM Atto594 Azide in 50 mM Tris buffer (pH 8) with 4 mM CuSO4 and 10 mM freshly prepared ascorbic acid. 4,6-Diamidino-2-phenylindole (DAPI) was added to visualize the nuclei. Coverslips were washed with 0.1% Triton in PBS and with PBS only and mounted with Aqua-Poly/Mount. Cells were imaged using a Zeiss LSM 700 Axio Imager Z2 upright microscope (Carl Zeiss Micro Imaging). The EU signal in the nuclei was quantified using ImageJ.

### Total extracts and immunoblotting

Cell pellets were lysed in denaturing lysis buffer (2% SDS, 1% NP-40, 150 mM NaCl, 50 mM Tris pH 7.5) with additional 50 U Benzonase® nuclease for 10 min at RT in rotation. Lysates were centrifuged 16,800 × $g$ for 10 min, and equal volumes of supernatant and 2× Laemmli-SDS sample buffer were heated at 98 °C for 5 min. Proteins were separated by SDS–PAGE using 4–12% Bis-Tris NuPAGE® gels with MOPS running buffer. Separated proteins were transferred onto PVDF membranes (0.45 µm) overnight at 4 °C, blocked in 5% BSA in PBS and probed with the appropriate primary antibodies (supplementary Data 11). Membranes were washed with PBS containing 0.05% Tween-20 and incubated with IRDye-conjugated secondary antibodies (supplementary Data 11). Proteins were visualized by the Odyssey® Imaging System.

### Immunoprecipitation

GFP-DDB2, CSA-GFP ectopically expressing cell lines and the CSA-mC KI cells were mock-treated or irradiated with 10 or 30 J m$^{-2}$ UV at different time points before cell collection. Cell pellets were prepared from three confluent 145-cm$^2$ dishes per condition for IP or MS. Cells were collected by trypsinization and pelleted in cold PBS by centrifugation for 5 mins. After one wash with cold PBS, cell pellets were stored at −80 °C until IP analysis. For IP, pellets were thawed on ice and lysed for 10 min on ice in HEPES buffer containing 30 mM HEPES pH 7.5; 130 mM NaCl; 1 mM MgCl$_2$; 0.5% Triton X-100; 1× EDTA-free Protease Inhibitor Cocktail. After 10 cycles of sonication using the Bioruptor Sonicator (15 sec on; 45 sec off) at 4 °C, 500 U Benzonase® nuclease was added, and samples were kept in rotation for 1–2 h at 4 °C. The insoluble fraction was pelleted at 16,800 g for 10 min at 4 °C, and the soluble fraction was applied for immunoprecipitation for 90 min at 4 °C, using 25 µl slurry GFP-Trap®A beads. Bound proteins were directly digested by trypsin for MS data independent analysis (DIA) or eluted with SDS–PAGE loading buffer and separated on 4–12% Bis-Tris NuPAGE® gels and processed for immunoblotting or for MS-data dependent analysis approach (DDA).

### Immunofluorescence

Immunofluorescence was carried out as previously described[70]. Cells were grown on 24-mm glass coverslips and fixed for 15 min in PBS with 3.7% FA. Subsequently, cells were permeabilized with 0.1% Triton X-100 in PBS and washed with PBS+ (0.15% BSA and 0.15% glycine in

PBS). Cells were incubated for 2 h at room temperature with rabbit anti-CSA, DDB2, GFP antibodies (supplementary Data 11) in PBS+. Thereafter, cells were washed with PBS+, 0.1% Triton and PBS+ before incubating for 2 h at room temperature with donkey anti-rabbit Alexa Fluor 488 or anti-rabbit Alexa Fluor 594 or anti-mouse Alexa Fluor 488 conjugated antibody (supplementary Data 11) and DAPI. Alexa Fluor 647 Phalloidin was used to detect actin. After washes with PBS+ and 0.1% Triton, coverslips were mounted with Aqua-Poly/Mount. Images were acquired with a Zeiss LSM700 Axio Imager Z2 upright microscope equipped with a ×63 Plan-apochromat 1.4 NA oil-immersion lens (Carl Zeiss Micro Imaging). The intensities were quantified using ImageJ.

## UV laser accumulation

Accumulation of proteins to UV laser-induced DNA damage was measured on a Leica SP8 confocal microscope (with LAS X software version 3.3.0.16799), coupled to a 4.5 mW pulsed (15 kHz) diode-pumped solid-state laser emitting at 266 nm (Rapp Opto Electronic, Hamburg GmbH; Supplementary). Cells, grown on quartz coverslips, were imaged and irradiated through an Ultrafluar quartz ×100/1.35 NA glycerol immersion lens (Carl Zeiss Micro Imaging Inc.) at 37 °C and 5% CO2. The resulting accumulation curves were corrected for background values and normalized to the relative fluorescence signal before local irradiation. After background correction, signals in the damaged and non-damaged areas of the nucleus were normalized to the average fluorescence levels of pre-damage conditions.

## FRAP

For FRAP, a Leica TCS SP8 microscope equipped with a ×40/1.25 NA HC PL APO oil-immersion lens (Leica Microsystems) was used. CSB-mC KI and CSA-mC KI cells were maintained at 37 °C and 5% CO$_2$ during imaging. Cells were seeded on glass coverslip two days prior to live imaging experiments and were treated with indicated UV doses and/or incubated with transcription inhibitors THZ1 (1 μM) 1 h before live cell imaging, or treated with CRL inhibitor NAE1i (1 μM) 30 min before FRAP analysis. A narrow strip of 512 × 16 spanning the nucleus was imaged at 400 Hz using a 488-m laser with a zoom of 8×. A total of 30 frames were measured to reach steady-state levels before photobleaching, followed by two frame 100% laser power. After photobleaching, the redistribution or recovery of fluorescence was measured with 200 ms frames until steady state was reached. Fluorescence intensity was background-corrected, normalized to the average of the last 30 pre-bleach frames and set to 100%. During one experiment for each condition at least 10 cells were measured. The immobile fraction ($F_{imm}$) was calculated as described in (REF) with the formula: $F_{imm} = 1 - (I_{final, treat} - I_{0, treat})/(I_{final, untr} - I_{0, treat})$.

## K-GG enrichment

Analysis of the global proteome and enrichment for diGly remnant-containing peptides using antibody-based enrichment was performed as described before[52]. Briefly, proteolytic peptides were fractionated using high pH reverse-phase (RP) chromatography. For the RP chromatography, a protein digest: stationary phase ratio of 1:50 was used and peptides were eluted in three fractions using increasing amounts of acetonitrile (7%, 13.5% and 50%). Fractions and flowthrough were subsequently dried to completeness by lyophilization. For immunoprecipitation of diGly peptides, ubiquitin remnant motif (K-ε-GG) antibodies coupled to beads (PTMscan) were used. After immunoprecipitation, the supernatant was stored for further global proteome analysis.

## Crosslinking, sample preparation and sequential digestion

The biochemically reconstituted complex consisting of recombinant UVSSA, USP7, CSB, CSA, DDB1, and DDA1 (20 μg), was immunoprecipitated using Pierce™ Anti-DYKDDDDK (anti-FLAG) Affinity Resin and re-suspended in a crosslinking buffer (20 mM HEPES, 20 mM NaCl, 5 mM MgCl$_2$, pH 7.8). Bissulfosuccinimidyl suberate (20 μg) was added to achieve a 1:1 (w/w) protein-to-crosslinker ratio, and the sample was crosslinked for 1 hour at room temperature with agitation. The reaction was quenched with excess ammonium bicarbonate (final concentration 20 mM) for 1 hour at room temperature. Following quenching, the supernatant was removed, and the resin-bound complex was subjected to on-bead digestion. The sample was re-suspended in elution buffer (8 M urea, 100 mM Tris, pH 8.0), reduced with dithiothreitol (final concentration 10 mM), alkylated with iodoacetamide (final concentration 50 mM), and diluted with 50 mM ammonium bicarbonate to achieve a final concentration of 1 M Urea. The eluate was collected and subjected to tryptic digestion at a protease-to-substrate ratio of 1:50 (w/w) overnight at 37 °C. Proteolysis was stopped with 10% (v/v) trifluoroacetic acid, and peptides were desalted using the StageTip method. Subsequently, the sample was eluted and divided into five aliquots before being evaporated in a vacuum concentrator. Sequential digestion was performed as follows:

1. For trypsin/AspN digestion: peptides were dissolved in 50 mM ammonium bicarbonate, and AspN was added to reach a 1:200 (w/w) protease-to-substrate ratio.

2. For trypsin/elastase digestion: peptides were dissolved in 50 mM Tris pH 8.8, and elastase was added to achieve a 1:50 (w/w) protease-to-substrate ratio.

3. For trypsin/GluC digestion: peptides were dissolved in 50 mM ammonium bicarbonate, and GluC was added to achieve a 1:200 (w/w) protease-to-substrate ratio.

4. For trypsin/ProAlanase digestion: peptides were dissolved in 10 mM HCl pH 2.0, and ProAlanase was added to reach a 1:200 (w/w) protease-to-substrate ratio.

All samples underwent overnight digestion at 37 °C. The reactions were stopped with 10% (v/v) trifluoroacetic acid. Finally, peptides were desalted by StageTipping and analyzed by LC-MS/MS.

## Mass spectrometry

**MS DATA-dependent analysis.** CSA-mClover protein complexes were pulled down from chromatin-enriched protein extracts with GFP-Trap®A beads as described previously[32]. Eluted proteins in Laemmli-SDS sample buffer were separated on 4–12% Bis-Tris NuPAGE® gels with MOPS running buffer and visualized with Coomassie. After cutting the gel lanes into 2-mm slices, the proteins were in-gel reduced with dithiothreitol, alkylated with iodoacetamide and digested with trypsin. Peptides were separated on a home-made 20 cm × 100 μm C18 column (BEH C18, 130 Å, 3.5 μm, Waters, Milford, MA, USA) after trapping on a nanoAcquity UPLC Symmetry C18 trapping column (Waters, 100 Å, 5 μm, 180 μm × 20 mm), using an EASY-nLC 1000 liquid-chromatograph. Subsequent MS analyses were performed on a Thermo Scientific Orbitrap Fusion™ Lumos Tribrid™ MS or an Orbitrap Eclipse™ Tribrid™ MS directly coupled to the EASY-nLC. All mass spectra were acquired in profile mode. The resolution in MS1 mode was set to 120,000 (AGC target: 4E5), the m/z range 350-1400. Fragmentation of precursors was performed in 2 s cycle time data-dependent mode by HCD with a precursor window of 1.6 m/z and a normalized collision energy of 30.0; MS2 spectra were recorded in the orbitrap at 30,000 resolution or in ion Trap as indicated in supplementary Data 10. Singly charged precursors were excluded from fragmentation and the dynamic exclusion was set to 60 seconds.

Raw MS data were analyzed with the MaxQuant software suite[44] as described previously[71,72]. The A false discovery rate (FDR) of 0.01 for proteins and peptides and a minimum peptide length of 7 amino acids were set. The Andromeda search engine was used to search the MS/MS spectra against the Uniprot database (taxonomy: Homo sapiens, release December 2022) concatenated with the reversed versions of all sequences. A maximum of two missed cleavages was allowed. The

peptide tolerance was set to 10 ppm and the fragment ion tolerance was set to 0.6 Da for HCD spectra. The enzyme specificity was set to trypsin and cysteine carbamidomethylation was set as a fixed modification. For K-GG enriched samples, the number of missed cleavages was raised to 3, while the variable modifications were set to oxidation (M) and GlyGly (K) and no fixed modification was selected. Both the PSM and protein FDR were set to 0.01. In case the identified peptides of two proteins were the same or the identified peptides of one protein included all peptides of another protein, these proteins were combined by MaxQuant and reported as one protein group. Before further statistical analysis, the 'proteingroups.txt' table was filtered for contaminants and reverse hits.

**MS DIA**. CSA-mClover protein complexes were pulled down from chromatin-enriched protein extracts with GFP-Trap®A beads as described previously[32]. Proteins were reduced with dithiothreitol, alkylated with iodoacetamide and digested with trypsin on beads. Peptides were separated on a home-made 20 cm × 100 μm C18 column (BEH C18, 130 Å, 3.5 μm, Waters, Milford, MA, USA) after trapping on a nanoAcquity UPLC Symmetry C18 trapping column (Waters, 100 Å, 5 μm, 180 μm × 20 mm), using an EASY-nLC 1000 liquid-chromatograph. Subsequent MS analyses were performed on Thermo Scientific Orbitrap Fusion™ Lumos Tribrid™ MS Orbitrap Eclipse™ Tribrid™ MS directly coupled to the EASY-nLC. All spectra were recorded at a resolution of 120,000 for full scans in the scan range from 350–1650 m/z. The maximum injection time was set to 50 ms (AGC target: 4E5). For MS2 acquisition, the mass range was set to 336–1391 m/z with dynamic isolation windows ranging from 7–82 m/z, with a window overlap of 1 m/z. The orbitrap resolution for MS2 scans was set to 30,000. The maximum injection time was at 54 ms (AGC target: 5E4; normalized AGC target: 100%).

DIA raw data files were analyzed with the Spectronaut Pulsar X software package (Biognosys, version 17.0.221202), using directDIA for DIA analysis including MaxLFQ as the LFQ method and Spectronaut IDPicker algorithm for protein inference. The Q-value cutoff at precursor and protein level was set to 0.01. All imputation of missing values was disabled.

**Crosslink-MS DATA**. The samples were analyzed using a Thermo Scientific reverse-phase EASY-nLC 1000 system coupled online to a Thermo Orbitrap Fusion Tribrid MS. Crosslinked peptides were separated using a gradient of buffer B (80% acetonitrile, 0.1% formic acid) comprising of the following steps: 7–15% gradient over 5 minutes, 15–35% over 214 minutes, 35–50% over 5 minutes, 50–95% over 1 minute, followed by 95% buffer B over 5 minutes. The total data acquisition time was 240 minutes. The flow rate was set to 200 nl/min. Only precursor ions with a charge state of 3 or greater were subjected to high-energy collision-induced dissociation (HCD) fragmentation. Mass spectra were recorded using a top20 acquisition mode. The raw data of crosslinked TC-NER complex were analyzed using MaxQuant software (version 2.1.4.0) with integrated MaxLynx and searched against the database comprising the amino acid sequences of the human TC-NER subunits (UniProt accession numbers: Q13216, Q03468, Q2YD98, Q9BW61, Q93009 and Q16531). Default parameters were applied, including carbamidomethylation of cysteine and oxidation of methionine set as fixed and variable modifications, respectively. BS3 was selected as the crosslinker. Four miscleavages were allowed for the tryptic digestion, while up to 11 miscleavages were permitted for the sequential proteolytic digests. Additionally, peak refinement was enabled, with peptide-spectrum match (PSM) false discovery rate (FDR) crosslink threshold set at 0.01. Parameters specifying the minimal peptide length and maximal peptide charge were defined as 6 and 8, respectively, with a maximum peptide mass set to 6000 Da. Subsequently, crosslink networks were visualized employing XINET and XIVIEW.

MS raw data and data for protein identification and quantification were submitted as supplementary tables to the ProteomeXchange Consortium via the PRIDE partner repository with the data identifier PXD045415 and PXD051638.

A roadmap for the raw files is included in the manuscript (supplementary Data 10), containing useful information to sort the data. In this roadmap, we indicate for each raw file the MS1 resolution, whether it was acquired on an ion trap or Orbitrap, the software used, the type of MS (SILAC or Free label), and whether it was acquired using DDA or DIA.

### RNA sequencing
**Library preparation, sequencing.** Total RNA was isolated from CSAKO, DDA1 and WT HCT116 cells (three independent experiments) with the RNeasy Lipid Tissue Mini Kit. RNA quantity and quality were evaluated using the NanoDrop 8000 spectrophotometer and Agilent 2100 Bioanalyzer, respectively. Single-read sequencing was performed in the experiment. Lexogen QuantSeq 3′ mRNA-seq V2 Library Prep kit with 12nt UDI was used for library preparation, including the UMI second strand synthesis kit, following the manufacturer instruction. Sequencing was performed on NextSeq 2000, using a 51/8/8 cycle setup to sequence the read 1, first index, and second index, respectively. 4 nucleotides were manually removed in the sample sheet from each index to ensure proper demultiplexing. This was managed in situ by DRAGEN suite version 07.021.624.3.10.12. Samples were on average sequenced to the depth of 0.75 gigabases.

**Bioinformatic analysis.** Reads were analyzed using FastQC v0.12.1 and MultiQC v1.18[73] to determine the overall quality of sequencing. Reads were then trimmed to remove adapter sequence leftovers and trimmed for quality using the BBTools suite. Afterwards, any reads aligning to human rRNA were also removed (bbsplit; U13369.1. Available online at: https://www.ncbi.nlm.nih.gov/nuccore/555853). STAR 2.7.11b[74] was used to align reads to genome build hg38 patch 13. QC metrics were extracted using GATK CollectRnaSeqMetrics[75]. Only primary aligned clusters that were part of the primary reference genome (contigs chr1-22, chrX, chrY,chr M) were then taken forward to be quantified on the exome and gene metafeature level using featureCounts v2.0.6[76].

**Differential gene expression.** EdgeR v4.0.2 was used for intersample normalization. Transcripts whose counts per million values did not exceed 1 in at least 2 samples were removed. Three differential expression experiments were then performed, one for each treatment. Treatment status was modelled as a dichotomous variable with no intersection using the glmQLFit function. Common biological coefficient of variation was calculated per experiment. Differentially expression was then carried out using the glmQLFTest. P values were corrected using the Benjamini-Hochberg procedure. Any genes with thus calculated FDR not surpassing 0.05 were deemed statistically significantly differentially expressed.

### Expression clones for recombinant proteins
All coding genes are full-length and of human origin. Except where noted, individual ORFs were cloned into pETNKI vectors by ligase-independent cloning[77]. The pAC8-CSA-Strep II clone was a gift from Nicolas Thomä[11]. The N-terminal 6×histidine tagged DDB1 gene was synthesized and codon-optimized for insect cell expression (gene synthesis services, Integrated DNA Technologies). The DDA1 construct was derived from GFP-DDA1 (gene synthesis services, GenScript), a TwinStrep-flag tag was introduced to the C-terminus. The CUL4A and RBX1 constructs[78,79] were obtained from Yue Xiong laboratory, both genes were fused in-frame to a N-terminal 6×histidine tag. The N-terminal 6×histidine tagged UVSSA gene was synthesized and codon-optimized for insect cell expression (gene synthesis services, Integrated DNA Technologies). The pFastBac-HA-CSB-His6 construct

was derived from Wim Vermeulen laboratory, the coding sequence contains an N-terminal HA tag and a C-terminal 6×histidine tag[80]. The codon-optimized pGEX-6p1-USP7 and pGEX-6p1-USP7[C223A] construct contained a codon optimized for bacterial expression and a GST tag was fused at the N-terminus[81,82]. The bacterial expression vectors pGEX-APPBP1-UBA3, pGEX-UBE2M and pGEX-NEDD8 were gifts from Brenda Schulman[83].

For protein complex co-expression in insect cells (CSA-DDB1-DDA1, CSA-DDB1, CUL4A-RBX1), biGBac polycistronic expression system was generated. The individual gene expression cassettes were amplified by PCR and integrated into a pBIG1a vector by Gibson assembly as described previously[84].

## CSA-DDB1-DDA1 and CSA-DDB1 purification
The CSA complexes with and without DDA1 were expressed in Sf9 insect cells and purified using an analogous procedure. Pellet from 2 L Sf9 culture was re-suspended in lysis buffer (20 mM HEPES pH 7.5, 150 mM NaCl, 5% glycerol (v/v), 0.1 mM EDTA, 0.5 mM TCEP, 30 mM imidazole). Cells were opened by sonication and the debris was removed by centrifugation at 53,340 × g for 30 min at 4 °C. Clarified lysate was loaded onto 5 ml Nickel-chelating sepharose and washed with 150 ml lysis buffer. The protein was eluted with lysis buffer containing 300 mM imidazole. The eluate was applied to a Resource Q column and then eluted with a 200–600 mM NaCl gradient. Peak fractions were collected, concentrated and injected into Superdex 200 16/600 column pre-equilibrated with SEC buffer (20 mM HEPES pH 7.5, 150 mM NaCl, 5% glycerol (v/v), 0.1 mM EDTA, 0.5 mM TCEP). The peak fractions were concentrated to around 5 mg/ml using an Amicon ultrafiltration device. Proteins were frozen in liquid nitrogen and stored at −80 °C.

## UVSSA purification
Pellet from 2 L Sf9 culture was re-suspended in high salt lysis buffer (20 mM HEPES pH 7.5, 500 mM NaCl, 5% glycerol (v/v), 0.1 mM EDTA, 0.5 mM TCEP, 30 mM imidazole). Cells were opened by sonication and the debris was removed by centrifugation at 53,340 × g for 30 min at 4 °C. Clarified lysate was loaded onto 5 ml Nickel-chelating sepharose and washed with 100 ml high salt lysis buffer and 50 ml low salt lysis buffer (20 mM HEPES pH 7.5, 150 mM NaCl, 5% glycerol (v/v), 0.1 mM EDTA, 0.5 mM TCEP, 30 mM imidazole). The protein was eluted with low salt lysis buffer containing 300 mM imidazole. The eluate was applied to a Resource S column and then eluted with a 150-450 mM NaCl gradient. Fractions containing UVSSA were collected and diluted two times with dilution buffer (20 mM HEPES pH 7.5, 5% glycerol (v/v), 0.1 mM EDTA, 0.5 mM TCEP). The diluted UVSSA was absorbed onto a 5 ml HiTrapQ column and eluted with a 200–500 mM NaCl gradient. To concentrate the protein, the peak fractions were collected and dialyzed overnight against storage buffer (20 mM HEPES pH 7.5, 300 mM NaCl, 50% glycerol (v/v), 0.1 mM EDTA, 0.5 mM TCEP). Protein was frozen in liquid nitrogen and stored at −80 °C.

## Preparation of K414 mono-ubiquitinated UVSSA
UVSSA can be in vitro mono-ubiquitinated by E2 enzyme UBE2E1 (UbcH6) in an E3-independent manner[19]. The UVSSA after HiTrapQ purification was used for large-scale preparation of mono-ubiquitinated UVSSA. 50 mM Bis-Tris-Propane pH 9.0, 0.5 μM UBA1, 16 μM UBE2E1, 20 μM ubiquitin, 10 mM MgCl2, 1 mM TCEP were added into UVSSA. The reaction was initiated by adding 5 mM ATP and incubate at room temperature for 30 min, and then the reaction was purified again with HiTrapQ column. The peak fractions were collected and dialyzed against storage buffer as described above.

## CSB purification
Pellet from 2 L Sf9 culture was re-suspended in high salt lysis buffer (20 mM HEPES pH 7.5, 500 mM NaCl, 5% glycerol (v/v), 0.1 mM EDTA,

0.5 mM TCEP, 30 mM imidazole). Cells were opened by sonication and the debris was removed by centrifugation at 53,340 × g for 30 min at 4 °C. Clarified lysate was loaded onto 5 ml Nickel-chelating sepharose and washed with 100 ml high salt lysis buffer and 50 ml low salt lysis buffer (20 mM HEPES pH 7.5, 150 mM NaCl, 5% glycerol (v/v), 0.1 mM EDTA, 0.5 mM TCEP, 30 mM imidazole). The protein was eluted with low salt lysis buffer containing 300 mM imidazole. The eluate was applied to a Heparin column and then eluted with a 150–1000 mM NaCl gradient. The peak fractions were concentrated and injected into a Superdex 200 16/600 column pre-equilibrated with SEC buffer (20 mM HEPES pH 7.5, 450 mM NaCl, 5% glycerol (v/v), 0.1 mM EDTA, 0.5 mM TCEP). The CSB fractions were concentrated to around 5 mg/ml. Aliquots were frozen in liquid nitrogen and stored at −80 °C.

## USP7 and USP7[C223A] purification
The wild-type USP7 and its C223A mutation were purified as previously described with minor modifications[81]. Pellet from 4 L E. coli Rosetta2(DE3) pLysS culture in TB medium was re-suspended in high salt lysis buffer (20 mM HEPES pH 7.5, 500 mM NaCl, 5% glycerol (v/v), 0.1 mM EDTA, 0.5 mM TCEP). Cells were opened by sonication and the debris was removed by centrifugation at 53,340 × g for 30 min at 4 °C. Clarified lysate was loaded onto 5 ml glutathione sepharose 4B resin and washed with 100 ml high salt lysis buffer and followed with 50 ml low salt buffer (20 mM HEPES pH 7.5, 150 mM NaCl, 5% glycerol (v/v), 0.1 mM EDTA, 0.5 mM TCEP). The protein was eluted with low salt buffer containing 20 mM reduced glutathione. The GST tag was removed by 3 C protease under dialysis against Q buffer (20 mM HEPES pH 7.5, 50 mM NaCl, 5% glycerol (v/v), 1 mM EDTA, 0.5 mM TCEP). The sample was then loaded onto a Resource Q column and eluted with a 100–450 mM NaCl gradient. The peak fractions were collected and injected into Superdex 200 16/600 column pre-equilibrated with SEC buffer (20 mM HEPES pH 7.5, 150 mM NaCl, 5% glycerol (v/v), 0.1 mM EDTA, 0.5 mM TCEP). The fractions of pure USP7 were concentrated to around 5–12 mg/ml using an ultrafiltration device. Proteins were frozen in liquid nitrogen and stored in −80 °C.

## Neddylated CUL4A-RBX1 purification
Pellet from 2 L Sf9 culture was re-suspended in lysis buffer (20 mM HEPES pH 7.5, 200 mM NaCl, 10% glycerol (v/v), 0.1 mM EDTA, 0.5 mM TCEP, 30 mM imidazole). Cells were opened by sonication and the debris was removed by centrifugation at 53,340 × g for 30 min at 4 °C. Clarified lysate was loaded onto 5 ml Nickel-chelating sepharose and washed with 150 ml lysis buffer. The protein was eluted with the same buffer containing 300 mM imidazole. The eluate was diluted to 120 mM NaCl and loaded onto a HiTrapSP column. Proteins were eluted with 120–500 mM NaCl gradient. The peak fractions were collected and treated with 3 C protease to remove the tags. The proteins were injected into a Superdex 200 16/600 column pre-equilibrated with SEC buffer (20 mM HEPES pH 7.5, 200 mM NaCl, 10% glycerol (v/v), 0.5 mM TCEP). The fractions of CUL4A-RBX1 were concentrated for in vitro neddylation reaction.

The in vitro neddylation was carried out in a 2 ml reaction containing 9 μM CUL4A-RBX1, 0.2 μM APPBP1-UBA3, 4 μM UBE2M, and 30 μM NEDD8 in 20 mM HEPES pH 7.5, 200 mM NaCl, 10% glycerol, 2 mM ATP, 5 mM MgCl2 and 0.5 mM TCEP. The reaction was incubated at room temperature for 30 minutes and terminated by adding EDTA to 50 mM. The neddylated CUL4A-RBX1 was further purified by cation ion exchange and size exclusion chromatography as described above.

## Cryo-EM sample preparation
UVSSA-Ub, USP7[C223A], and CSA-DDB1-DDA1 were mixed in 1:1:1 ratio and incubated on ice for 10 min. The complex was crosslinked by GraFix[85] in which the sample was applied to a glycerol gradient containing 20 mM HEPES pH 7.5, 150 mM NaCl, 0.5 mM TCEP, 10-30% glycerol and 0-0.12% glutaraldehyde. The gradient was

ultracentrifuged at 90,000 × g for 16 hours at 4 °C. Fractions containing the crosslinked complex were pooled and exchanged to the same buffer without glycerol using Zeba spin desalting column (Thermo Fisher Scientific). To prepare cryo-EM grids, 3 μl of cross-linked sample was applied on a Quantifoil R1.2/1.3 Cu 300 grid pre-coated with graphene oxide. The grid was blotted for 2.5 s and frozen in liquid ethane using Vitrobot Mk IV plunge freezer operating at 4 °C and 100% humidity.

### Cryo-EM data collection and processing
The micrographs were acquired on FEI Titan Krios 300 kV electron microscope (NeCEN, the Netherlands) with a K3 detector (Gatan) and an energy filter (Gatan) with slit width of 20 eV. Automated data collection was using EPU (ThermoFisher Scientific) and the movies were acquired in a magnification of ×81,000 (1.09 Å/pix) with a dose of 60 e −/Å over 50 frames.

The initial data processing was carried out in cryoSPARC[86]. Particles were picked using TOPAZ[87]. The TOPAZ picking model was trained using manual picked particles from subset of micrographs. After 2D classification clean-up and consensus 3D refinement, the coordinates were imported to Relion3.0[88]. In Relion, micrographs were motion-corrected by MotionCor2 and the contrast transfer function was estimated by CTFFIND-4.1[89,90]. Due to structural heterogeneity, iterations of particle subtraction and focused classification were conducted. The final 3D reconstruction maps were sharpened by DeepEMhancer[91] (for map 2 and map 3) or EM-GAN[92] (for map 1). For illustration purpose, the three maps were combined using the combine-focused-maps tool in Phenix[93]. The details of the processing are described in Supplementary Figs. 3–4.

### Model building
Model building and refinement were done in Chimera, Phenix and Coot[93–95]. For CSA-DDB1 (BPA and BPC), the available model protein data bank (PDB) code 7OO3[21] was rigid body fitted to the map and manual adjusted in Coot[94]. The missing loops were built with the guidance of AlphaFold2 model[40]. For DDA1 the crystal structure the PDB code 6PAI[36] and the AlphaFold2 model[40] were used as the initial reference, and the C-terminal helix was rebuilt in Coot. The core structure containing CSA-DDB1(BPA/BPC)-DDA1 was real space refined in Phenix and PDB-REDO[93,96]. The DDB1 BPB domain was rigid body fitted with available crystal structure PDB code 3EI3[11] with minor adjustment. The UVSSA VHS domain was fitted with Alpha-Fold2 model from residue 1–150. The pixel size of the cryo-EM maps was recalibrated in Chimera by calculating the cross-correlation between the experimental map and a calculated map generated by the CSA model from AlphaFold2. The models were built with the maps in pixel size 1.057 Å. Structure figures were generated using ChimeraX[97].

### Differential scanning fluorometry
The thermostability of CSA-DDB1-DDA1, CSA-DDB1-DDA1[1–52] and CSA-DDB1 were analyzed by differential scanning fluorometry (DSF). Samples were diluted to 2 μM in reaction buffer (20 mM HEPES pH 7.5, 150 mM NaCl, 5% glycerol, 0.5 mM TCEP) and applied on a Prometheus NT.48 using the standard capillaries. The intrinsic tryptophan fluorescence 350/330 nm ratio was measured in a 1 °C/min temperature gradient from 20 to 90 °C.

### Ubiquitination assays
For analysis of E3-dependent UVSSA ubiquitination, 10 μl reactions were set up on ice containing 1.5 μM UVSSA, 0.2 μM UBA1, 1 μM UBE2E1 (UbcH6), 20 μM ubiquitin and indicated amount of neddylated CRL4A[CSA] in 20 mM HEPES pH 7.5, 100 mM NaCl, 2 mM ATP, 5 mM MgCl2 and 0.5 mM TCEP. Reactions were initiated by adding ATP and incubated at 30 °C for 20 minutes. Reactions were terminated with SDS loading buffer. Proteins were separated on Bolt 8% Bis-Tris Plus gels (Thermo Fisher Scientific) in MOPS buffer and stained with Coomassie blue.

For CSB ubiquitination, analogous reactions were set up except using 1 μM CSB as substrate and 1 μM UBE2D3 (UbcH5c) as E2 enzyme. Proteins were separated on Bolt 4–12% Bis-Tris Plus gels in MOPS buffer and stained with Coomassie blue.

### Statistical analysis
Mean values, as well as each individual value and S.D. or S.E.M. error bars, are shown for each experiment. The number of samples analyzed per experiment is reported in the respective figure legends. Multiple t-tests (unpaired, two-tailed) were used to determine statistical significance between groups, followed by multiple comparison corrections with the Holm-Sidak method. For analysis of Fig. 3C, D, Fig. 4B, C, Supplementary Fig. 1D, Supplementary Fig. 7B, F, G, Supplementary Fig. 8A, Supplementary Fig. 9B, C, a nested t test (two-tailed) was performed with significance levels set to 0.05. All analyses were performed using Graph Pad Prism version 8.2.1 for Windows (GraphPad Software, La Jolla, CA, USA). p values ≤ 0.05 were considered significant, otherwise as non-significant. The networks were generated through the use of QIAGEN IPA[53] (QIAGEN Inc., https://digitalinsights.qiagen.com/IPA).

### Reporting summary
Further information on research design is available in the Nature Portfolio Reporting Summary linked to this article.

## Data availability
MS raw data and data for protein identification and quantification have been deposited to the ProteomeXchange Consortium via the PRIDE partner repository and are available with the data identifier PXD045415 or PXD051638. RNA sequencing raw data are available at the SRA database (https://www.ncbi.nlm.nih.gov/sra) with the data identifier PRJNA1103704. Cryo-EM data were submitted to the Electron Microscopy Data Bank (EMDB) [https://www.ebi.ac.uk/emdb] and to the PDB [https://www.rcsb.org] with the data identifier EMD-18377, EMD-18378, EMD-18380, EMD-18413, and 8QH5. Source data are provided with this paper.

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

## Acknowledgements
This work was supported by the CW-TOP grant (714.017.003 to W.V., T.K.S., and M.V.), ALW grant (ALWOP.143 to J.H.J.H.), EMBO long-term fellowship (ALTF 1159-2019 to S.H.L.), Oncode Institute (partly financed by the Dutch Cancer Society to W.V., T.K.S., M.V., and J.H.J.H.), National Institute of Health (NIH)/National Institute of Ageing (NIA) (P01 AG017242 to J.H.J.H.), ZonMw Memorabel (project ID 733050810 to J.H.J.H.), European Research Council Advanced Grant Dam2Age, DFG (German Research Foundation)—FOR 5504 (496650118 to J.H.J.H.), and the European Joint Project on Rare Diseases RD20-113, acronym TC-NER to J.H.J.H.). Research at the Netherlands Cancer Institute is supported by institutional grants from the Dutch Cancer Society and the Dutch Ministry of Health, Welfare, and Sport. The content is solely the responsibility of the authors and does not necessarily represent the official views of the National Institutes of Health. We thank the NeCEN staff for their help in data collection. We thank Benita Quist, Clarissa Polidoro Pontalti and Stanley Van for their technical help in the experiments. We thank Vid Prijatelj, Pascal Arp, and Mila Jhamai for their help with RNA sequencing.

## Author contributions
W.V., A.P., and D.A.L.S. conceived the research. D.A.L.S. generated knockout cells, constructs, and stable cell lines, performed RRS and clonogenic survival assays, all live-cell imaging experiments, PCR, and western blot analysis to validate knockouts. S.H.L. generated expression clones for recombinant proteins, purified recombinant proteins, prepared, collected, and processed the Cryo-EM samples, and performed the DSF. K.W.K. performed XL-MS experiments and analyzed the MS samples. A.F.T., A.H., and M.A. generated knockout cells. K.B. performed Ubiquitin enrichment experiments for MS and analyzed the MS samples. C.G.H. performed RRS assays. M.v.d.V. performed Co-IP experiments for western blot analysis. A.P. performed Co-IP experiments for western blot analysis, Co-IP experiments for MS, and RRS assays. T.S., M.V., W.V., and A.P. supervised the project. D.A.L.S., S.H.L., T.K.S., W.V., and A.P. draft the paper. D.A.L.S., S.H.L., K.W.K., H.v.A., A.C.O.V., J.H.J.H., J.A.M., H.L., J.A.A.D., M.V., T.S., T.O., W.V., and A.P. contributed to conceptualization, and reviewed and edited the draft.

## Competing interests
The authors declare no competing interests.

## Additional information

[1]Department of Molecular Genetics, Erasmus MC Cancer Institute, Erasmus University Medical Center, 3015 GD Rotterdam, The Netherlands. [2]Division of Biochemistry and Oncode institute, Netherlands Cancer Institute, Plesmanlaan 121, 1066CX Amsterdam, The Netherlands. [3]Department of Molecular Biology, Faculty of Science, Radboud Institute for Molecular Life Sciences (RIMLS), Oncode Institute, Radboud University Nijmegen, 6525 GA Nijmegen, the Netherlands. [4]Proteomics Center, Erasmus University Medical Center, 3015 GD Rotterdam, The Netherlands. [5]Department of Molecular Genetics, Erasmus MC Cancer Institute, Oncode Institute, Erasmus University Medical Center, 3015 GD Rotterdam, The Netherlands. [6]Department of Human Genetics, Leiden University Medical Center, 2333 ZC Leiden, The Netherlands. [7]Department of Cell and Chemical Biology, Leiden University Medical Center, 2333 ZC Leiden, The Netherlands. [8]University Hospital of Cologne, CECAD Forschungszentrum, Institute for Genome Stability in Aging and Disease, Joseph Stelzmann Strasse 26, 50931 Köln, Germany. [9]Princess Maxima Center for Pediatric Oncology, Oncode Institute, Heidelberglaan 25, 3584 CS Utrecht, the Netherlands. [10]Division of Molecular Genetics and Oncode institute, The Netherlands Cancer Institute, Plesmanlaan 121, Amsterdam 1066 CX, the Netherlands. [11]Department of Genetics, Research Institute of Environmental Medicine (RIeM), Nagoya University, Nagoya, Japan. [12]Department of Human Genetics and Molecular Biology, Graduate School of Medicine, Nagoya University, Nagoya, Japan. [13]Present address: Max Planck Institute of Molecular Physiology, Otto-Hahn-Straße 11, 44227 Dortmund, Germany. [14]Present address: Cancer Science Institute of Singapore, National University of Singapore, 14 Medical Drive, Singapore 117599, Singapore. [15]These authors contributed equally: Diana A. Llerena Schiffmacher, Shun-Hsiao Lee. ✉e-mail: w.vermeulen@erasmusmc.nl; a.pines@erasmusmc.nl

