## [Peer Review File · Nature Communications]

The small CRL4^{CSA} ubiquitin-ligase component DDA1 regulates transcription-coupled repair dynamicsREVIEWER COMMENTS

Reviewer #1 (Remarks to the Author):

The authors used mClover-tagged CSA to determine the CSA interactome by performing a pull-down followed by mass spectrometry analysis. Results were validated by cryo-EM and by functional assays. Overall, the experiments appear to be well-designed and controlled. Specifically, the authors did a good job of demonstrating that the mClover tag did not impact the function of CSA in the knock-in cell lines. Given the nature of the study, this is an important point to demonstrate. To me, this work is suitable for publication after minor revisions. They are as follows:

1. Unless I misread, the authors intersperse CSA-mClover and CSA-mC throughout the text when referring to labeled CSA. If this is a mistake, it should be corrected.
2. As per the methods section, MS2 spectra were acquired at 30,000 resolution in the DDA experiments. However, database search parameters in the experimental section and the experimental description in PRIDE sounded indicated that the spectra were acquired at low resolution. I downloaded a data file and checked, and MS2 were acquired in the ion trap rather than the Orbitrap. The authors need to correct the DDA description to reflect this. Aside from the error in the DDA description, the data acquisition and analysis descriptions are very good.
3. The authors are to be commended for making their data available; more scientists could follow their example. However, within the ProteomeXchange upload, it is difficult to sort out what all the data files and searches are. It would be nice if the authors included a roadmap of some sort for interested readers. The submission appears to be incomplete, but it is difficult to tell.

Reviewer #2 (Remarks to the Author):

Alex Pines from the Rotterdam group, along with a network comprising 6 laboratories and 20 researchers, is dedicated to further dissecting the mechanism of TCR when elongating RNAPII encounters DNA damage.

Utilizing highly sophisticated technologies, they initially identified a protein called DDA1 that co-elutes with a GFP-CSA and appears to be a part of the TCR complex. DDA1 interacts with CSA, a member of the CRL4 ubiquitin ligase complex previously identified by multiple laboratories which could potentially be involved in the ubiquitination of RNAPII.

Following the acquisition of several suitable cell lines, including HCT/CSA-Ki, HCT/DDA1, and others, they convincingly demonstrated a partnership between DDA1 and components of the CRL4/CSA complex. They conducted several rigorous controls, which strengthened the reliability of their data. Furthermore, they revealed that this isolated complex (obtained through immunoprecipitation and subsequent mass spectrometry) also includes CSB, TRIC, and COP9 components, all of which are involved in the TCR mechanism.

Their next step involved tentatively investigating the structure of a UVSSA/CSA/DDB1/DDA1 complex. However, I have reservations about the validity of this complex's structure. Cryo-electron microscopy (cryo-EM) can provide high-resolution structural information, but it is crucial to confirm that the captured complex structure accurately represents the natural interactions. In comparison with findings from Cramer's lab, I am curious if the presence of RNAPII and DNA in the complex could help determine the relevance and potential function of DDA1 in the TCR process. In this scenario, although I acknowledge the group's efforts in conducting this structural study, I question the added value and whether the interactions identified here are intermediate, "artificial," or natural, such as those in the TCR complex.

The group's demonstration that DDA1 knockout (KO) did not significantly alter UV survival (Figure 3C) but revealed a clear RRS defect. This might suggest a more transcriptional role rather than a TCR function. High UV doses could potentially affect other functions, such as protein modifications. They also investigated the role of DDA1 in some ubiquitination processes. However, they did not observe significant overall changes (including within the RPB1 subunit of RNAPII) in the UV-activated pathway between wild-type (WT) and KO cells (Figure 5E and 5F).

Despite my appreciation for the quality of their various experiments and the numerous convincing controls, I perceive these data as very preliminary. Furthermore, given that they have identified DDA1 and established several cell lines, it raises the question of why they did not perform RNA-seq and ChIP-seq, using antibodies targeting various TCR (ubiquitinated) components, to assess the global impact of DDA1 in regulating transcription, possibly in a gene-specific manner. They also could have sequentially determined the prerequisites for the incorporation of DDA1 into the RNAPII and/or TCR complex.

Reviewer #3 (Remarks to the Author):

The manuscript by Schiffmacher et al. investigates the role DDA1 in regulating transcription coupled nucleotide excision repair (TC-NER). TC-NER is an important repair mechanism for the removal of transcription-blocking DNA lesions (TBLs). The focus on DDA1 is within the context of its interaction with Cullin 4 multisubunit E3 ligase complex, where CSA is the substrate receptor component. Whilst not an expert in the field of DNA damage, it would appear the discovery that DDA1 forms an integral component of the CUL4CSA complex is novel. However, I'm not sure if this would be a surprise to the direct field because DDA1 is known to associate with many CUL4 complexes. It's unfortunate that despite the extensive and well-carried-out experimentation, the data do not underscore a clear functional role within the Cul4CSA ligase, specifically in the context of DNA repair.

A wealth of data is reported but the manuscript would benefit from further data demonstrating a critical function for DDA1. Insight at a molecular level would also further strengthen the manuscript.

Major points

It might be possible to validate the cryoEM interaction between CSA and DDA1 by introducing mutations within CSA or DDA1 at the interface and see if this abolishes the thermal stabilisation. The weak density at this interface brings the significance of this interaction into question, and the reviewer is wondering whether this is relevant and if complex formation is solely driven by the previously known DDA1 interaction with DDB1.

As DDA1 forms complexes with multiple CUL4 complexes, it is not clear if the modest defect in RRS is attributable to it no longer interacting with CUL4CSA or whether this is a pleiotropic effect.

While the data presented in figure 3 clearly show that DDA1 is required for efficient RRS, there is no evidence this is because of its association with CSA within the context of a complex with DDB1. Would it be possible to perform a rescue experiment with WT DDA1 and a DDA1 mutant predicted to impair binding to CSA (guided by cryoEM structure)?

Data in Figure 4 suggesting a role for DDA1 in nuclear localisation is convincing, but experiments with NLS tags in the absence of DDA1 do not phenocopy, suggesting that the DDA1 has an additional function.

The DIA data showing enhanced complex formation between clover-CSA and UVSSA and CSB RNAPII etc, should be validated by IP immunoblotting.

diGly proteomics has not yielded much insight. Although a different substrate profile was detected without DNA damage in DDA1 KO cells, there is no evidence this is due to loss of DDA1 binding to CUL4CSA.

Minor points

TFIIH should be defined before its first introduction and its relevance discussed in the introduction.

Figure 2 c and e would benefit from annotation to make the significance readily apparent.

What is the molecular basis for stabilised binding of CSB to stalled RNAPII?

Reviewer #1 (Remarks to the Author):

The authors used mClover-tagged CSA to determine the CSA interactome by performing a pull-down followed by mass spectrometry analysis. Results were validated by cryo-EM and by functional assays. Overall, the experiments appear to be well-designed and controlled. Specifically, the authors did a good job of demonstrating that the mClover tag did not impact the function of CSA in the knock-in cell lines. Given the nature of the study, this is an important point to demonstrate. To me, this work is suitable for publication after minor revisions. They are as follows:

We thank this reviewer for her/his appreciation on the quality of our manuscript and the value of the study for the DNA repair research community.

1. Unless I misread, the authors intersperse CSA-mClover and CSA-mC throughout the text when referring to labeled CSA. If this is a mistake, it should be corrected.

We apologize for the mistake. Throughout the text, we have replaced 'CSA-mClover knock-in' with 'CSA-mC KI'.

2. As per the methods section, MS2 spectra were acquired at 30,000 resolution in the DDA experiments. However, database search parameters in the experimental section and the experimental description in PRIDE sounded indicated that the spectra were acquired at low resolution. I downloaded a data file and checked, and MS2 were acquired in the ion trap rather than the Orbitrap. The authors need to correct the DDA description to reflect this. Aside from the error in the DDA description, the data acquisition and analysis descriptions are very good.

We apologize for the inconvenience. We have adjusted the material and methods section and to avoid any problem related to the MS data, we have generated a roadmap for the raw files (supplementary Table 10). In this roadmap, we indicate for each raw file the MS1 resolution, whether it was acquired on an ion trap or Orbitrap, the software used, the type of MS (SILAC or Free label), and whether it was acquired using DDA or DIA.

3. The authors are to be commended for making their data available; more scientists could follow their example. However, within the ProteomeXchange upload, it is difficult to sort out what all the data files and searches are. It would be nice if the authors included a roadmap of some sort for interested readers. The submission appears to be incomplete, but it is difficult to tell.

We agree with the reviewer's observation that 'it is difficult to sort out what all the data files and searches are.' As mentioned above, we have prepared a roadmap, an excel file (supplementary Table 10), containing useful information to sort the data. We would like to thank the reviewer for the comment, and we are confident that the data are now easier to sort and find for interested readers.

Reviewer #2 (Remarks to the Author):

Alex Pines from the Rotterdam group, along with a network comprising 6 laboratories and 20 researchers, is dedicated to further dissecting the mechanism of TCR when elongating RNAPII encounters DNA damage. Utilizing highly sophisticated technologies, they initially identified a protein called DDA1 that co-elutes with a GFP-CSA and appears to be a part of the TCR complex. DDA1 interacts with CSA, a member of the CRL4 ubiquitin ligase complex previously identified by multiple laboratories which could potentially be involved in the ubiquitination of RNAPII.

Following the acquisition of several suitable cell lines, including HCT/CSA-Ki, HCT/DDA1, and others, they convincingly demonstrated a partnership between DDA1 and components of the CRL4/CSA complex. They conducted several rigorous controls, which strengthened the reliability of their data. Furthermore, they revealed that this isolated complex (obtained through immunoprecipitation and subsequent mass spectrometry) also includes CSB, TRIC, and COP9 components, all of which are involved in the TCR mechanism.

We thank this reviewer for her/his appreciation on the quality of our manuscript.

Their next step involved tentatively investigating the structure of a UVSSA/CSA/DDB1/DDA1 complex. However, I have reservations about the validity of this complex's structure. Cryo-electron

microscopy (cryo-EM) can provide high-resolution structural information, but it is crucial to confirm that the captured complex structure accurately represents the natural interactions. In comparison with findings from Cramer's lab, I am curious if the presence of RNAPII and DNA in the complex could help determine the relevance and potential function of DDA1 in the TCR process. In this scenario, although I acknowledge the group's efforts in conducting this structural study, I question the added value and whether the interactions identified here are intermediate, "artificial," or natural, such as those in the TCR complex.

We agree that a cryo-EM structure of RNAPII with all known TC-NER factors together would bring additional insight and potentially help to more precisely determine the function of DDA1. We thank the reviewer for pointing this out, therefore we superimposed the DDA1-containing complex from this study onto the recently published Pol II-ELOF1-TCR complex (PDB 8B3D, in which DDA1 is not present) (new Supplementary Figure 5 A). To further strengthen the validity of CSA-DDA1 interaction, we performed AlphaFold prediction of CSA-DDB1-DDA1 (Supplementary figure 4C-D) and compared it with the cryo-EM obtained structure. The five predicted models are internally consistent, show high confidence and are consistent with the cryo-EM structure in this study. The residue 54-77 of DDA1 are predicted as an α -helix (The "C-terminal helix" interacting with DCAFs). In the cryo-EM structure, we observe this helix only till residue 61, the rest is invisible most likely due to high flexibility. Comparing the CSA-DDA1 interaction to the crystal structure of DCAF15-DDA1 from (PDB 6UD7), in which the corresponding helix has larger interface with DCAF15, the helix is visible till residue 72. We argue that the C-terminal helix of DDA1 will become flexible when losing the contact with DCAF. Additionally, AlphaFold prediction of the extended TC-NER complex (Supplementary figure 5B), confirmed this (see also rebuttal to reviewer #3). The model also shows that DDA1 can integrate into the complex without clashing with other subunits, indicating that the proposed DDA1 conformation is highly likely within the context of the whole complex. Interestingly, the DDA1 C-terminal helix will pass through a cavity created by CSA, DDB1 and CSB.

In the article we added a paragraph (results section, DDA1 is a component of the CSA/DDB1 complex) and a supplementary figure as follows: "Recent structural studies have elucidated how CRL4^{CSA} is assembled within the TC-NER complex in a context with RNA Pol II and CSB^{21,42}. To understand how DDA1 is arranged in this complex, we superimposed our structure into the Pol II-ELOF1-TCR complex⁴². In the model, the C-terminal helix of DDA1 passes through a cavity surrounded by CSA, DDB1 and CSB (Supplementary Figure 5 A-B). The extension of the C-terminal helix is exposed near the ATPase domain of CSB."

"To examine and/or confirm whether DDA1 would interact with CSA, DDB1 and CSB, we applied cross-link Mass Spectrometry (XL-MS). We assembled the CSA-DDB1-DDA1 complex, together with UVSSA, USP7 and CSB. The complex was chemically cross-linked and the bound proteins were digested into covalently cross-linked peptides. Identification of cross-linked peptides, by mass spectrometry and analyzed using MaxQuant⁴⁴ software with integrated MaxLynx⁴⁵ revealed residues in close spatial proximity. We identified 140 unique, high confidence residue cross-links in total (Supplementary Figure 5 C and Supplementary Table 3). Importantly, we observed 30 different cross-links between DDA1 and either CSA, DDB1, or CSB, involving DDA1 residues Lys13, Lys26, Lys51, Lys65, Lys66 and Lys70 (Supplementary Table 3). Remarkable, no crosslink peptides were found between DDA1 and UVSSA or USP7, as expected, considering the observed spatial distance between these proteins. Although this analysis does not provide information about specific residues that mediate the interaction, the location of lysine 51, 65, 66 and 70 residues in DDA1 and their association to the outer regions of the β -propeller blades made up by the WD40 domain of CSA and C-terminal domain CSB are in line with the cryo-EM comparison in which DDA1 is located within a cavity created by CSA, DDB1, and CSB (supplementary Figure 5 A-B). The XL-MS support the cryo-EM data and strengthen the role of DDA1 as a component into the TC-NER complex"

The group's demonstration that DDA1 knockout (KO) did not significantly alter UV survival (Figure 3C) but revealed a clear RRS defect. This might suggest a more transcriptional role rather than a TCR function. High UV doses could potentially affect other functions, such as protein modifications. They also investigated the role of DDA1 in some ubiquitination processes. However, they did not observe significant overall changes (including within the RPB1 subunit of RNAPII) in the UV-

activated pathway between wild-type (WT) and KO cells (Figure 5E and 5F). Despite my appreciation for the quality of their various experiments and the numerous convincing controls, I perceive these data as very preliminary. Furthermore, given that they have identified DDA1 and established several cell lines, it raises the question of why they did not perform RNA-seq and ChIP-seq, using antibodies targeting various TCR (ubiquitinated) components, to assess the global impact of DDA1 in regulating transcription, possibly in a gene-specific manner.

We would like to highlight that the UV irradiation of 10 J/m² is considered a low dose compared to high UV doses typically used in most TC-NER studies, which can reach up to 500 J/m² (PMID: 30344095). In the context of this work, 10 J/m² UV has been utilized as the highest dose, although it falls within a range that we consider relatively low irradiation.

The reviewer raised an important question regarding whether DDA1 functions solely as a transcription-regulating factor rather than being essential for TC-NER. We thank the reviewer for this concern and give us the opportunity to resolve the issue. We have included several pieces of evidence demonstrating that DDA1 is not a transcription factor:

- Our existing proteomic data indicate no significant downregulation of relevant TC-NER proteins and the overall global protein levels are comparable to the WT (Supplementary Table 6). To highlight this point further, we have generated a new illustration correlating protein expression between WT and DDA1KO cells (supplementary Figure 7 D-F). This analysis confirmed that the protein level is not significantly altered in both conditions (CSA KO and DDA1 KO), further arguing that if transcription would be slightly altered it certainly does not affect DDR protein expression, thereby ruling out that the observed TC-NER defect would be derived from a transcription defect.
- RNA synthesis, measured by EU incorporation, did not show any difference between WT and DDA1 KO cells, further emphasizing that there is no global impact on transcription by the absence of DDA1. This information is already in the paper (Figure 3 B, D on RRS), but due to common normalization procedures this is not revealed. We have added a new figure to show directly that there is no overall impact on transcription in the absence of DDA1 (supplementary Figure 7 A).
- Additionally, we have performed RNA sequencing experiments and included them in the manuscript, as suggested by the reviewer. The RNA-seq data confirmed that transcription is not altered in DDA1 KO conditions. Together with the aforementioned experiments, this new data strongly indicates that DDA1 is not a transcription factor (supplementary Figure 7 B-C). We have included these new data in the manuscript (see result section, paragraph "DDA1 is required for transcription recovery following DNA damage")

They also could have sequentially determined the prerequisites for the incorporation of DDA1 into the RNAPII and/or TCR complex.

We do not really understand this comment, since the DDA1 protein is part of CRL4-CSA complex (as we demonstrated in this manuscript). Incorporation of DDA1 into the TCR complex is thus through the established (PMID: 32355176) loading of CRL4-CSA to lesion-stalled RNAPII-CSB. The significance of DDA1 in TC-NER is clearly demonstrated in Figure 6, showing its role in TC-NER progression and complex disassembly with live cell imaging studies (Figure 6D), in IP-experiments (Figure 6E) and by MS (Supplementary figure 9 A, B). However, to further support these data we have performed FRAP experiments using the CSB-mClover KI cell line in absence or presence of DDA1. In line with the CSA mobility a significant fraction of CSB-mClover molecules remained immobilized in DDA1 KO cells, whereas in WT cells the mobility was fully recovered to the same level as in undamaged cells (supplementary Figure 15 C). The data have been included in the manuscript.

Reviewer #3 (Remarks to the Author):

The manuscript by Schiffmacher et al. investigates the role DDA1 in regulating transcription coupled nucleotide excision repair (TC-NER). TC-NER is an important repair mechanism for the removal of transcription-blocking DNA lesions (TBLs). The focus on DDA1 is within the context of its interaction with Cullin 4 multisubunit E3 ligase complex, where CSA is the substrate receptor component. Whilst

not an expert in the field of DNA damage, it would appear the discovery that DDA1 forms an integral component of the CUL4CSA complex is novel. However, I'm not sure if this would be a surprise to the direct field because DDA1 is known to associate with many CUL4 complexes. It's unfortunate that despite the extensive and well-carried-out experimentation, the data do not underscore a clear functional role within the Cul4CSA ligase, specifically in the context of DNA repair.

A wealth of data is reported but the manuscript would benefit from further data demonstrating a critical function for DDA1. Insight at a molecular level would also further strengthen the manuscript. We sincerely appreciate the reviewer's recognition of the quality of our manuscript. Indeed, DDA1 is known to associate with several CUL4-DDB1-RBX complexes. However, its association with the CUL4-CSA complex (what we demonstrated in this manuscript) and its role in TC-NER were not known. We also showed that this association is in part specific, as not all CUL4 complexes rely on this association. We showed that DDA1 is not part of the CUL4-DDB2 complex (Fig.1C, Supplementary Figure 2) and not required for the nuclear targeting of this structurally similar complex (Fig.4A,B) that functions in GG-NER. We were, however, somewhat surprised to read that despite the '*wealth of data*' further experimental evidence is requested to demonstrate the functional role of DDA in the context of DNA repair.

We would like to emphasize that our work presents several experimentally-derived evidences regarding the role of DDA1 in TC-NER, which was previously never considered as a TC-NER factor, including:

-We identified DDA1 as a novel player in TC-NER by Mass Spectrometry (MS)-based interaction-proteomics.

-Cryo-electron microscopy analysis showed DDA1 as an integral component of the CSA-containing Cullin-based ubiquitin-ligase complex (CUL4CSA). With the new data added (refinement of cryo-EM data, superposition onto existing RNAPII-TC-NER complexes and XL-MS), this interaction is further confirmed.

-DDA1 emerged as a key player in enabling transcription recovery after UV-irradiation and protects cells against UV-induced damage.

-Live cell imaging studies, cellular fractionation and IP, and MS revealed that DDA1 facilitates the dynamic turnover of TC-NER and is important in regulating the progression of DNA repair.

Major points

It might be possible to validate the cryoEM interaction between CSA and DDA1 by introducing mutations within CSA or DDA1 at the interface and see if this abolishes the thermal stabilisation. The weak density at this interface brings the significance of this interaction into question, and the reviewer is wondering whether this is relevant and if complex formation is solely driven by the previously known DDA1 interaction with DDB1.

It should be noted that DDA1 association to CUL4 could not be solely driven by its interaction with DDB1, since we showed that DDA1 is not part of CUL4^{DDB2} complex (Figure 1C, Supplementary figure 2) and that its absence does not affect its subcellular localization as it does to CUL4^{CSA} (Figure 4).

However, to address the concern from the reviewer, we conducted the following novel experiments:

1. Thermal stability assay of the complex with DDA1 truncation.
 2. Improve the cryo-EM map by using an alternative sharpening approach.
 3. Comparison to AlphaFold prediction of the CSA-DDB1-DDA1 complex.
-
1. To test whether the CSA-DDA1 interaction has an effect on thermal stability, we purified the complex with DDA1 C-terminal truncation (CSA-DDB1-DDA1¹⁻⁵²) in which the C-terminal helix interacting with CSA is deleted. The result showed that the truncated complex has the same melting temperature as the full-length complex, indicating that the thermal stabilizing effect of DDA1 is mainly contributed by the part interacting with DDB1. This result is expected as the interaction between CSA-DDA1 is rather limited, and such weak interaction may not be recapitulated by thermal stability assay. Nevertheless, the increase stabilization of DDB1 could help to maintain the integrity of the whole E3 ligase, although the effect is not directly through protecting CSA.

To clarify the effect of DDA1, we modified the text and included the result in Supplementary Figure 4.

2. We agree that the provided cryo-EM map in the original figures do not provide sufficient clarity on the specific interaction of DDA1 with CSA. We believe that this is due to the weak interaction and limited interface. To improve the map quality, we revisited the data and tested alternative map sharpening tools. This approach is often useful to improve the map at low resolution. We used the newly developed program EM-GAN (Maddhuri Venkata Subramaniya, S. R. et al. *Bioinformatics*, 2023) for sharpening the Map 1 (Supplementary figure 4), which was reconstructed with more particles and showed better quality of CSA-DDB1-DDA1. The new map provides a better-quality map of the CSA-DDA1 interface. We are convinced that this weak interaction is valid.

We built the atomic model according to Map 1 and changed Figure 2D for better presentation. The newly generated map and coordinate were updated accordingly on EMDB and PDB.

3. To strengthen the validity of CSA-DDA1 interaction, we performed AlphaFold prediction of CSA-DDB1-DDA1 (Supplementary Figure 4 C-E). The five predicted models are internally consistent, show high confidence and are consistent with the cryo-EM structure in this study. The residue 54-77 of DDA1 are predicted as an α -helix (The “C-terminal helix” interacting with DCAFs). In the cryo-EM structure, we observe this helix only till residue 61, the rest is invisible most likely due to high flexibility. Comparing to the DCAF15-DDA1 interaction from crystal structure (PDB 6UD7), in which the corresponding helix has larger interface with DCAF15, the helix is visible till residue 72. We argue that the C-terminal helix of DDA1 will become flexible when losing the contact with DCAF.

In the article we added a paragraph to describe the AlphaFold analysis (Material and Methods, and result section, paragraph “DDA1 is a component of the CSA/DDB1 complex”) and provide figures in Supplementary Figure 4 C-E.

In addition, as suggested by reviewer#2, we also superimposed the DDA1-containing CRL4^{CSA} complex from this study onto the recently published Pol II-ELOF1-TCR complex (Supplementary Figure 5B). The model shows that DDA1 can integrate into the complex without clashing with other subunits, indicating that this DDA1 conformation is possible within the context of the whole complex. Interestingly, the DDA1 C-terminal helix will pass through a cavity created by CSA, DDB1 and CSB (supplementary Figure 5 A-B). These intriguing findings, supported by XL-MS data (supplementary Figure 5 C), imply that the C-terminal region of DDA1 extends beyond its canonical association within the CRL^{CSA} complex. Instead, it appears to interact with additional components of the TC-NER machinery, such as CSB.

As DDA1 forms complexes with multiple CUL4 complexes, it is not clear if the modest defect in RRS is attributable to it no longer interacting with CUL4CSA or whether this is a pleiotropic effect.

DDA1-KO cells showed a clear RRS defect (Figure 3 B) after 10 J/m² UV irradiation, similar to CSA and CSB KO cells in which the TC-NER is not functional. The reviewer raised an important question regarding whether this is a pleiotropic effect as DDA1 is a component of multiple CUL4 complexes. To address the concern from the reviewer, we have determined the overall protein profile, the overall RNA synthesis capacity and the gene expression by RNA sequencing in DDA1KO cells and compared it to isogenic DDA1 expressing cells (see reviewer#2, supplementary Figure 7 A-F and supplementary table 3 and 6). Our findings suggest that DDA1 does not modulate the gene expression or protein levels of DNA repair proteins, suggesting that the observed TC-NER-deficiency is specific. Importantly, RRS and colony survival assays, cryo-EM, MS and live imaging experiments show a direct role of DDA1 in TC-NER.

While the data presented in figure 3 clearly show that DDA1 is required for efficient RRS, there is no evidence this is because of its association with CSA within the context of a complex with DDB1.

Would it be possible to perform a rescue experiment with WT DDA1 and a DDA1 mutant predicted to impair binding to CSA (guided by cryoEM structure)?

Despite utilizing new software that enables a more precise delineation of the DDB1/DDA1/CSA CryoEM structure and a better description of the region of DDA1 that interacts with CSA, we were not able to define specific amino acids. This challenge arises from the intrinsic mobility and transient nature of the interaction between DDA1 and CSA. As a solution, we deleted the interacting C-terminal part of DDA1 implicated in this interaction. The functional impact has been studied using Crisp-CAS9 system to generate DDA1 deletions in cells and ectopically express the truncated version of DDA1 (Delta, Δ) in DDA1KO cells. The data shown in the new supplementary Figure 6 C-G confirm that the C-terminal part of DDA1 is required for functional TC-NER. The results were also discussed in the result and discussion sections.

Data in Figure 4 suggesting a role for DDA1 in nuclear localisation is convincing, but experiments with NLS tags in the absence of DDA1 do not phenocopy, suggesting that the DDA1 has an additional function.

We are confused by this comment from reviewer #3, as this is exactly the statement in our main text, *i.e.*: “These experiments strongly suggest that the TC-NER defect in DDA1KO cells is associated to another, thus far unidentified, molecular mechanism rather than to a reduced nuclear protein level of CSA.” It is not clear to us, what is meant or requested with this remark.

The DIA data showing enhanced complex formation between clover-CSA and UVSSA and CSB RNAPII etc, should be validated by IP immunoblotting.

As requested, we validated the MS data by IP immunoblotting and included the new data, which confirm the MS results, in the revised manuscript, in the manuscript (supplementary Figure 10 A-C)

diGly proteomics has not yielded much insight. Although a different substrate profile was detected without DNA damage in DDA1 KO cells, there is no evidence this is due to loss of DDA1 binding to CUL4CSA.

Our ubiquitin profile data indicate that under non-damaging conditions, the same pathways were affected in the absence of either DDA1 or CSA (figure 6 A). These pathways range from DDR-associated signaling, mRNA processing, translational mechanism, mitochondrial function and protein folding/stability processes. These data may suggest that CUL4^{CSA}, including DDA1, is activated by the presence of endogenous DNA damage, and that the absence of CSA or DDA1 affect the same processes. These findings strongly suggest that these proteins are intimately related.

Following the reviewer's comment, we conducted an in-depth comparison of the ubiquitination profiles in the absence of DNA damaging agents. We specifically examined ubiquitin sites that were quantified in all mock experiments, totaling 2947 unique ubiquitin sites. Unfortunately, several ubiquitin sites were found to be inconsistent among all experiments and were therefore not considered in the comparison. We applied a threshold filter of 2-FC (-1 Log_2), resulting in the identification of 41 ubiquitin sites that were downregulated in CSA KO cells. Interestingly, 10 of these ubiquitin sites were shared with DDA1KO, and even 18 (45%) when the cutoff was set at -0.5 Log_2 . Notably, this includes the K1268 ubiquitin site of RNAPII, which is specifically involved in regulating TC-NER. We included a new figure (Supplementary Figure 14 D) and added the data of this analysis in supplementary Table 9, to illustrate these findings.

Minor points

TFIIH should be defined before its first introduction and its relevance discussed in the introduction. We have provided references for readers to deepen their understanding of the functionality and crucial importance of TFIIH. However, we have defined TFIIH before its first introduction and its relevance in the introduction section.

Figure 2 c and e would benefit from annotation to make the significance readily apparent.

We thank the reviewer for the suggestion. We modified the figure for better presentation.

What is the molecular basis for stabilised binding of CSB to stalled RNAPII?

To provide additional evidence we conducted further experiments to validate and expand our findings. The new data unequivocally demonstrate the importance of the C-terminal region of DDA1 in TC-NER performance. DDA1 resides within a pocket formed by CSA, DDB1, and CSB, and its role does not involve transcription factor activity. These intriguing findings suggest that the C-terminal region of DDA1 extends beyond its canonical association within the CRL^{CSA} complex, likely modulating the protein network of the CRL4^{CSA} complex, beyond the ligase complex (i.e. towards CSB) and its ubiquitination targets (i.e K1268 RNAPII). Consequently, we observed a defective or inefficient TC-NER process, highlighted by the stabilized binding of CSB to stalled RNA polymerase II. These insights significantly enhance our understanding of the intricate mechanisms underlying TC-NER regulation.

For the new XL-MS data, please find below the Submission details:

Project Name: DDA1, a novel factor in transcription-coupled repair, modulates CRL4^{CSA} dynamics at DNA damage-stalled RNA polymerase II

Project accession: PXD051638

Reviewer account details:

Username: reviewer_pxd051638@ebi.ac.uk

Password: czaUrDSL

REVIEWERS' COMMENTS

Reviewer #1 (Remarks to the Author):

The authors have addressed my concerns, and I am happy to recommend this paper for publication.

Cheryl Lichti

Reviewer #2 (Remarks to the Author):

The authors did as much as possible to answer to my points. Therefore I may consider that such work can be published. The most important point being the evidence that DDA1 is a DNA repair factor and not a transcription factor. Of course it could be worthwhile to check its role in some in vitro assay. I hope and guess that this point will be under investigation in their lab.

Reviewer #3 (Remarks to the Author):

The authors have made great efforts to revise the manuscript. It is much stronger for it and I recommend publication.

Reviewer #1 (Remarks to the Author):

The authors have addressed my concerns, and I am happy to recommend this paper for publication.

Reviewer #2 (Remarks to the Author):

The authors did as much as possible to answer to my points.

Therefore I may consider that such work can be published. The most important point being the evidence that DDA1 is a DNA repair factor and not a transcription factor. Of course it could be worthwhile to check its role in some in vitro assay. I hope and guess that this point will be under investigation in their lab.

Reviewer #3 (Remarks to the Author):

The authors have made great efforts to revise the manuscript. It is much stronger for it and I recommend publication.

We are grateful to the reviewers for their insightful feedback, constructive comments, and suggestions, as well as their unanimous recognition of the manuscript's quality. We must admit that the revisions have considerably improved the work.